# Simulating the atmospheric $CO_2$ concentration across the heterogeneous landscape of Denmark using a coupled atmosphere-biosphere mesoscale model system

Anne Sofie Lansø[1,a], Thomas Luke Smallman[2,3], Jesper Heile Christensen[1,4], Mathew Williams[2,3], Kim Pilegaard[5], Lise-Lotte Sørensen[4], and Camilla Geels[1]

[1]Department of Environmental Science, Aarhus University, Frederiksborgvej 399, 4000 Roskilde, Denmark
[2]School of GeoSciences, University of Edinburgh, Edinburgh, EH9 3JN, UK
[3]National Centre for Earth Observation, University of Edinburgh, Edinburgh, EH9 3JN, UK
[4]Arctic Research Centre (ARC), Department of Bioscience, Aarhus University, Ny Munkegade 114, 8000 Aarhus, Denmark
[5]Department of Environmental Engineering, Technical University of Denmark (DTU), Bygningstorvet 115, 2800 Kgs. Lyngby, Denmark
[a]Now at: Laboratoire des Sciences du Climat et l'Environnement, LSCE/IPSL, CEA-CNRS-UVSQ, Université Paris-Saclay, 91191 Gif-sur-Yvette, France

*Correspondence to:* Anne Sofie Lansø (anne-sofie.lanso@lsce.ipsl.fr)

**Abstract.** Although coastal regions only amount to 7% of the global oceans, their contribution to the global oceanic air-sea $CO_2$ exchange is in-proportionally larger with fluxes in some estuaries being similar in magnitude to terrestrial surface fluxes of $CO_2$.

Across a heterogeneous surface consisting of a coastal marginal sea with esturine properties and varied land mosaics, the surface fluxes of $CO_2$ from both marine areas and terrestrial surfaces were investigated in this study together with their impact in atmospheric $CO_2$ concentrations by the usage of a high-resolution modelling framework. The simulated terrestrial fluxes across the study region of Denmark experienced an east-west gradient corresponding to the distribution of the landcover classification, their biological activity and the urbanized areas. Annually, the Danish terrestrial surface had an uptake of approximately -7000 GgC yr$^{-1}$. While the marine fluxes from the North Sea and the Danish inner waters annually were smaller with about -1,800 GgC yr$^{-1}$ and -1,300 GgC yr$^{-1}$, their sizes are comparable to annual terrestrial fluxes from individual landcover classifications in the study region, and hence not negligible. The contribution of terrestrial surfaces fluxes were easily detectable on both simulated and measured concentrations of atmospheric $CO_2$ at the only tall tower site in the study region. Although, the tower is positioned next to Roskilde fjord, the local marine impact was not distinguishable in the simulated concentrations. But the regional impact from the Danish inner waters and the Baltic sea increased the atmospheric concentration by up to 0.5 ppm during the winter months.

# 1 Introduction

Understanding the natural processes responsible for absorbing just over half of the anthropogenic carbon emitted to the atmosphere, will help decipher future climatic pathways. During the last decade, the ocean and the biosphere are annually estimated to take up $2.4 \pm 0.5$ PgC yr$^{-1}$ and $3.0 \pm 0.8$ PgC yr$^{-1}$ of the $9.4 \pm 0.5$ PgC yr$^{-1}$ anthropogenic carbon emitted to the atmosphere (Le Quéré et al., 2018). The heterogeneity and the dynamics of the surface complicates such estimates.

Biosphere models of various complexity have been developed to spatially simulate surface fluxes of $CO_2$ but future estimates of the land uptake are bound with large uncertainties (Friedlingstein et al., 2014) that can be attributed to model structural uncertainties, uncertain observations and lack of model bench-marking (Cox et al., 2013; Luo et al., 2012; Lovenduski and Bonan, 2017). To have the best chance of accurately predicting the future evolution of the carbon cycle, and its implications for our climate, it is important to minimise the uncertainties that exist presently (Carslaw et al., 2018). Enhanced knowledge and a better process understanding in ecological theory and modelling could potentially reduce the model structural uncertainties (Lovenduski and Bonan, 2017) which together with improvements in spatial surface representation could minimise the current uncertainties.

Studying surface exchanges of $CO_2$ on regional to local scale can be accomplished with mesoscale atmospheric transport models. Their resolution is in the range from 2 km to 20 km and their advantage is their capability to get a better processes understanding of both atmospheric and surface exchange mechanisms in order to improve the link between observations and models at all scales, i.e. for both mesoscale, regional and global models (Ahmadov et al., 2007). The higher spatial resolution of mesoscale models allows for a better representation of atmospheric flows and for a more detailed surface description, which in particular for heterogeneous areas is necessary. In previous mesoscale model studies, biosphere models have been coupled to the mesoscale atmospheric models ranging in their complexity from simple diagnostic (Sarrat et al., 2007b; Ahmadov et al., 2007, 2009) to mechanistic process based biosphere models (Tolk et al., 2009; Ter Maat et al., 2010; Smallman et al., 2014; Uebel et al., 2017). The modelled $CO_2$ concentrations and surface fluxes from mesoscale model systems compare better with observations than global model systems (Ahmadov et al., 2009). The atmospheric impact on surface processes related to the ecosystem's sensitivity and $CO_2$ exchange can be examined in greater details (Tolk et al., 2009) and tall towers footprints can be studied more concisely (Smallman et al., 2014).

Heterogeneity can also be considerable in coastal oceans, and like terrestrial surface fluxes, the high spatiotemporal variability leads to large uncertainties in estimates of coastal air-sea $CO_2$ fluxes (Cai, 2011; Laruelle et al., 2013). Coastal seas play an important role in the carbon cycle facilitating lateral transport of carbon from land to open oceans, but almost 20 % of the carbon entering estuaries are released to the atmosphere, while 17 % of the carbon inputs to coastal shelves comes from atmospheric exchange (Regnier et al., 2013). The air-sea $CO_2$ exchange is in general numerical larger for estuaries than shelf seas (Chen et al., 2013; Laruelle et al., 2010, 2014), and can for estuaries annually be as large as 1,958 gC m$^{-2}$ yr$^{-1}$, while continental shelf seas have fluxes in the range of -154 gC/m2/yr to 180 gC m$^{-2}$ yr$^{-1}$ (Chen et al., 2013). The large spatial and temporal heterogeneity of coastal ocean adds to the large uncertainty related to the annual estimates of the air-sea $CO_2$ exchange (Regnier et al., 2013). The observed high spatial and temporal variability (Kuss et al., 2006; Leinweber et al., 2009;

Vandemark et al., 2011; Norman et al., 2013; Mørk et al., 2016) are not always included in marine models (Omstedt et al., 2009; Gypens et al., 2011; Kuznetsov and Neumann, 2013; Gustafsson et al., 2015; Valsala and Murtugudde, 2015), let alone taken into account in atmospheric mesoscale systems simulating $CO_2$ (Sarrat et al., 2007a; Geels et al., 2007; Law et al., 2008; Tolk et al., 2009; Broquet et al., 2011; Kretschmer et al., 2014). But a recent study found that short-term variability in the partial pressure of surface water $CO_2$ ($pCO_2$) substantially can affect simulated annual fluxes in certain coastal areas – in their case the Baltic Sea (Lansø et al., 2017). Moreover, direct Eddy Covariance (EC) measurements in the Baltic Sea have shown that upwelling events with rapid changes of $pCO_2$ greatly increase the air-sea $CO_2$ exchange (Kuss et al., 2006; Rutgersson et al., 2009; Norman et al., 2013)

In this study we aim to simulate surface exchanges of $CO_2$ at a high spatial-temporal resolution across a region, neighboring the Baltic Sea, alternating between land and coastal sea together with mesoscale atmospheric transport. A newly developed mesoscale modelling system is used to assess and understand the dynamics and relative importance of the marine and terrestrial $CO_2$ fluxes. The Danish Eulerian Hemispheric Model (DEHM) forms the basis of the framework, while the mechanistic biospheric Soil-Plant-Atmosphere model (SPA) is dynamically coupled to the atmospheric model. Both models are driven by methodological data from the Weather Research and Forecasting model (WRF). The air-sea $CO_2$ exchange is simulated at a high temporal resolution with the best applicable surface fields of $pCO_2$ for the Danish marine areas. Tall tower observations are used to evaluate the simulated atmospheric concentrations of $CO_2$.

Section 2 is dedicated to describe the study region, which is followed by a detailed description and evaluate of the atmospheric and biospheric model components of the developed model system in Sect. 3. Section 4 contains results, while discussion and conclusion follow in Sect. 5 and 6.

## 2   Study area

The study area comprises of Denmark, a country that is characterised by a mainland (Jutland) and many smaller islands, all containing varied land mosaic of urban, forest and agricultural areas. With more than 7,300 km of coastline encircling approximately 43,000 $km^2$ of land, many land-sea borders are found throughout the country adding to the complexity (Fig. 1). Denmark is positioned in the transition zone between the Baltic Sea, a marginal coastal sea with low salinity, and the North Sea, a continental shelf sea. Bordering the Baltic Sea, the Danish inner waters are rich on nutrients and organic material (Kuliński and Pempkowiak, 2011). This fosters high biological activity in spring and summer lowering surface water $pCO_2$ allowing for uptake of atmospheric $CO_2$. In winter, mineralisation increases $pCO_2$ (Wesslander et al., 2010), and outgassing of $CO_2$ to the atmosphere takes place. The North Sea is a persistent sink of atmospheric $CO_2$, where a continental shelf-sea pump efficiently removes $pCO_2$ from the surface water and transport it to the North Atlantic Ocean (Thomas et al., 2004). This study uses definition of the Danish exclusive economic zone (EEZ) to estimate the Danish air-sea $CO_2$ exchange, as the coastal state (in this case Denmark) has the right to explore, exploit and manage all resources found within its EEZ (United Nations Chapter XXI: Law of the Sea, 1984). The Danish EEZ is approximately 105,000 $km^2$.

A tiling approach with the seven most common biospheric landcover classification were selected for the current study including deciduous forest (3,348 $km^2$), evergreen forest (1,870 $km^2$), winter barley (1,211 $km^2$), winter wheat and other winter crops (9,269 $km^2$), spring barley and other spring crops ( 5,368 $km^2$), grassland (6,924 $km^2$) and agricultural other (3,909 $km^2$), but excluding urbanised areas. The agricultural other landcover classification includes all agricultural that does not classify as cereals, and as such contains root crops, fruits, corn, hedgerows and agricultural 'undefined'. This classification corresponds to the actual crop distribution of 2011 (Jepsen and Levin, 2013).

## 2.1 Observations of atmospheric $CO_2$

One tall tower is found within the study area on the eastern inner shore of Roskilde Fjord. Here atmospheric continuous measurements have been conducted at the Risø campus tower site (55°42′N, 12°05′E) during 2013 and 2014. The tower is located on small hill 6.5 m above sea level (Sogachev and Dellwik, 2017). Roskilde Fjord is a narrow micro-tidal estuary 40 km long with a surface area of 123 $km^2$, a mean depth of 3 m and found in the sector 200°- 360°relative to the Risø campus tower (Mørk et al., 2016). The city of Roskilde with around 50,000 inhabitants is positioned approximately 5 km southwest of the site, while Copenhagen lies 20 km towards east.

The tall tower continuous measurements of atmospheric $CO_2$ concentrations at Risø campus tower were carried out by the use of a Picarro G1301 placed in a heated building. The inlet was 118 m above the surface and the tube flow-rate was 5 slpm. The Picarro was new and calibrated by the factory. The calibration was checked by a standard gas of 1000 ppm $CO_2$ in atmospheric air (Air Liquide). During the measurement period from the middle of 2013 to the end of 2014, the instrument showed no other drift than the general increase in the global atmospheric concentration.

## 3 Model setup

The model framework used in the present study consists of two models; DEHM and SPA. A coupling between the two was made for the inner most nest of DEHM in order to simulate the exchange of $CO_2$ between the atmosphere and terrestrial biosphere at a high temporal (1 hour) and spatial resolution (5.6 km $\times$ 5.6 km) for the area of Denmark.

## 3.1 DEHM

DEHM is an atmospheric chemical transport model covering the Northern Hemisphere with a polar stereographic projection true at 60°N. Originally, developed to study sulphur and sulphate (Christensen, 1997), the DEHM model now contains 58 chemical species and nine groups of particular matter (Brandt et al., 2012). This adaptable model has been used to study atmospheric mercury (Christensen et al., 2004), persistent organic pollutants (Hansen et al., 2004), biogenic volatile organic compounds influence on air quality (Zare et al., 2014), emission and transport of pollen (Skjøth et al., 2007), ammonia and nitrogen deposition (Geels et al., 2012a, b) and atmospheric $CO_2$ (Geels et al., 2002, 2004, 2007; Lansø et al., 2015). The $CO_2$ version of DEHM was used in the present study. DEHM has 29 vertical levels distributed from the surface to the 100 hPa surface with approximately 10 levels in the boundary layer. Horizontally, DEHM has 96 $\times$ 96 grid points, which through its

nesting capabilities increases in resolution from 150 km × 150 km in the main domain to 50 km × 50 km, 16.7 km × 16.7 km and 5.6 km × 5.6 km in the three nests. The two-way nesting replaces the concentrations in the coarser grids by the values from the finer grids.

### 3.1.1 Surface fluxes in DEHM

Anthropogenic emissions of $CO_2$, wild fire emissions and optimized biospheric fluxes from NOAAs ESRL Carbon Tracker system (Peters et al., 2007) version CT2015 were used as inputs to DEHM. Their resolution is $1° \times 1°$ with updated values every third hour. Similarly, CT2015 three-hourly mole fractions of $CO_2$ were read in as boundaries conditions at the lateral boundaries of the main domain.

Hourly anthropogenic emissions on a 10 km × 10 km grid from the Institute of Energy Economics and the rational Use of Energy (IER, Pregger et al. (2007)) were applied for Europe instead of emissions from CT2015. Furthermore, these are for the area of Denmark substituted by hourly anthropogenic emissions with an even higher spatial resolution of 1 km × 1 km (Plejdrup and Gyldenkærne, 2011). As the European and Danish emission inventories were from 2005 and 2011, respectively, the emissions were scaled to annual national total $CO_2$ emission of fossil fuel and cement production conducted by EDGAR (Olivier et al., 2014), in order to include the yearly variability in national anthropogenic $CO_2$ emissions.

### 3.1.2 Air-sea $CO_2$ exchange

The exchange of $CO_2$ between the atmosphere and the ocean, $F_{CO_2}$, was calculated by $F_{CO_2} = Kk_{660}\Delta pCO_2$, where $K$ is solubility of $CO_2$ calculated as in Weiss (1974), $k_{660}$ is the transfer velocity of $CO_2$ normalised to a Schmidt number of 660 at $20°$C, and $\Delta pCO_2$ is the difference in partial pressure of $CO_2$ between the surface water and the overlying atmosphere. The transfer velocity parameterisation $k = 0.266u_{10}^2$, where $u_{10}^2$ is the wind speed at 10 m, determined by Ho et al. (2006) has been found to match Danish fjord systems (Mørk et al., 2016), and was applied in the current study. Surface values of marine $pCO_2$ were described by a combination of the open ocean surface water climatology of $pCO_2$ by Takahashi et al. (2014), and the climatology developed by Lansø et al. (2015, 2017) for the Baltic Sea and Danish waters. Furthermore, short-term temporal variability was accounted for in the surface water $pCO_2$ by imposing monthly mean diurnal cycles onto the monthly climatologies following the method described in Lansø et al. (2017).

### 3.1.3 Meteorological drivers

The necessary meteorological parameters for DEHM were simulated by the WRF (Skamarock et al., 2008), nudged by six hourly ERA-Interim meteorology (Dee et al., 2011) for the period 2008 to 2014, and were also used as initial and boundary conditions. In WRF the Noah Land Surface Model, Eta similarity surface layer and the Mellor-Yamada-Janjic boundary layer scheme were chosen to simulate surface and boundary layer dynamics. The CAM scheme was used for long and short wave radiation, the WRF Single-Moment 5-class Microphysics scheme was applied for micro-physical processes, and the Kain Fritsch scheme for cumulus parametrisation (Skamarock et al., 2008). In WRF the same nests as in DEHM were chosen,

and the meteorological outputs were saved every hour. To get the sub-hourly values that match the time step in DEHM, a temporal interpolation was conducted between the hourly time steps when DEHM was reading the hourly meteorological data. Furthermore, a correction of the horizontal wind speed was conducted in DEHM to ensure mass conservation and compliance with surface pressure (Bregman et al., 2003).

### 3.1.4 Evaluation of meteorological drivers

We did not have access to measurements from official meteorological observational sites. Thus, for the evaluation of the meteorological drivers, measurements from different types of monitoring sites were used comprising of three air pollution monitoring sites, three FLUXNET sites and three sites from the Danish Hydrological observatory and exploratorium (HOBE) (Table 1).

Wind directions, investigated by comparing wind roses made from WRF outputs and measurements, were at most sites reasonably captured by WRF (see supplement Fig. S1-S3). At several sites the frequency of wind directions from west were overestimated by WRF, mainly on the expense of southern winds. However, the opposite was the case at Aarhus, where the effect of street canyons likely was causing higher occurrences from due west in the observed wind directions. The wind velocities were in general overestimated by WRF with an average of 1.1 ms$^{-1}$ with the greatest differences at the same sites experiencing most problems in reproducing the observed wind direction patterns (Fig. 2). Moreover, at the Risø campus tower site the wind velocities were underestimated.

Only one site had available surface pressure measurements and high correlation of $R^2 = 0.99$ was obtained with the simulation surface pressures, indicating that WRF were capable of reproducing the actual pressure system across the study region (Fig. S4). Comparisons of wind velocities and wind rose likewise indicate that WRF captured the general atmospheric flow patterns, albeit the overestimated wind velocities might induce too quick atmospheric mixing. The simulated mixing layer heights have previously been evaluated, and although the diurnal boundary layer dynamics were reproduced together with the rectifier effect, problems with accurately modelling the night time boundary layer was observed, possibly overestimating night time surface concentrations of $CO_2$ (Lansø, 2016). Moreover, long-range transport and boundary conditions of atmospheric $CO_2$ concentrations have previously been shown to be captured by the model system across Northern Europe, in where the current study area is positioned, using observations from Mace Head, Pallas, Westerland, the oil and gas platform F3, Lutjewad and Östergarnsholm (Lansø et al., 2015).

Evaluating the meteorological variable also important for the biospheric model component, the surface temperature showed high correlation with $R^2$ above 0.93 for all sites (Fig. 3). The total shortwave incoming radiation (R$_{in}$) mirrors the measured R$_{in}$ from the three HOBE sites, but is during summer overestimated by WRF (Fig. S5). The values of photosynthetic active radiation (PAR) passed to SPA from DEHM, might thus be overestimated as PAR is proportional to R$_{in}$. However, in SPA there is a cap on the limiting carboxylation rate in the calculation of the photosynthesis, and the effect of the overestimated PAR can thus be limited. Precipitation was lacking from all sites, but the annual accumulated modelled precipitation at the nine sites follows the country wide annual estimates (Cappelan et al., 2018), however, with higher values for the westernmost sites, since many frontal systems enters Denmark from the west (Fig. S6).

## 3.2 SPA

The SPA model is a mechanistic terrestrial biosphere model (Williams et al., 1996, 2001). SPA has a high vertical resolution with up to 10 canopy layers (Williams et al., 1996) allowing for variation in the vertical profile of photosynthetic parameters and multi-layer turbulence (Smallman et al., 2013). Within the soil up to 20 soil layers can be simulated (Williams et al., 2001). The

5 radiative transfer scheme estimates the distribution of direct and diffuse radiation, and sunlit and shaded leaf areas (Williams et al., 1998). SPA uses the mechanistic Farquhar model (Farquhar and von Caemmerer, 1982) of leaf level photosynthesis, and the Penman-Monteith model to represent leaf level transpiration (Jones, 1992). Photosynthesis and transpiration are coupled via a mechanistic model of stomatal conductance, where stomatal opening is adjusted to maximise carbon uptake per unit nitrogen within hydraulic limitations, determined by a minimum leaf water potential tolerance, to prevent cavitation.

Ecosystem carbon cycling and phenology is determined by a simple carbon cycle model (DALEC, Williams et al. (2005), which is directly coupled into SPA. DALEC simulates carbon stocks in foliage, fine roots, wood (branches, stem and coarse roots), litter (foliage and fine root) and soil organic matter (including coarse woody debris). Photosynthate is allocated to autotrophic respiration and living biomass via fixed fractions, while turnover of carbon pools is governed by first order kinetics. In addition, when simulating crops, a storage organ (i.e. the crop yield) and dead, but still standing, foliage pools are added

influencing both radiative transfer and turbulent exchange (Sus et al., 2010).

SPA has been extensively validated against site observations from temperate forests (Williams et al., 1996, 2001), temperate arable agriculture (Sus et al., 2010) and Arctic tundra (Williams et al., 2000). SPA has more recently been coupled into the Weather Research and Forecasting (WRF) model (Skamarock et al., 2008), the resulting WRF-SPA model was used in multi-annual simulations over the United Kingdom and assessed against surface fluxes of $CO_2$, $H_2O$ and heat, and atmospheric

observations of $CO_2$ from aircraft and a tall tower (Smallman et al., 2013, 2014).

SPA needs vegetation and soil input parameters. Initial soil carbon stock estimates were obtained from the Regridded Harmonized World Soil Database (Wieder, 2014). The vegetation inputs and plant traits for SPA were partly taken from previous parameter sets used in SPA, but also from the Plant Trait Database (TRY, Kattge et al. (2011)), and from literature (Penning de Vries et al., 1989; Wullschleger, 1993). As these parameters and plant traits were determined at various sites that not necessar-

25 ily corresponds to Danish conditions, a calibration of the vegetation inputs to SPA was conducted for Danish Eddy Covariance (EC) flux sites (Table 2). Only data from five sites were available and these were divided in two sets – one for calibration (all available observations before 2013), the other validation (all available observations from 2013 and 2014).

In the inner most nest of DEHM for the area of Denmark, a coupling was made between DEHM and SPA. Thus, the coarser optimized biospheric fluxes from CT2015 were for Denmark replaced by hourly SPA simulated $CO_2$ fluxes. With this change,

the spatial resolution for the biosphere fluxes was increased from $1° \times 1°$ to $5.6$ km $\times 5.6$ km allowing for a better representation of the Danish surface, and hence also the biospheric fluxes.

On an hourly basis, DEHM provides atmospheric $CO_2$ concentrations and the meteorological drivers obtained from WRF to SPA, while SPA each hour returns net ecosystem exchange (NEE) to DEHM.

### 3.2.1 SPA calibration

The calibration was conducted by selecting a set of inputs parameters (plant traits, carbon stocks etc.) and for each parameter five values within a realistic range were chosen. Next, 200 SPA simulations with randomly chosen parameter values were conducted. These results were statistically evaluated against observations of NEE from the different flux sites, with the aim of selecting the parameter combination with the lowest root mean square error (RMSE) in combination with highest correlation that captured the observed variability and onset of the growing season. However, it was not always possible to have all these conditions satisfied (see e.g. Fig. S7). Based on this random parameter testing, it was possible to choose the best set of realistic vegetation input parameters that could improve the model performance at the Danish sites. The best found vegetation parameters values corresponded in some cases to the values already applied in SPA for the given land cover.

### 3.2.2 SPA evaluation

Comparing to observations of NEE, SPA was, in general, able to capture the phenology and seasonal cycle throughout the entire simulation period (Fig. 4). Correlations and RMSEs between the model and the independent data from the validation period (Fig. 4) likewise indicate a good model performance. At Sorø both variability, as inferred from the standard deviations, the amplitude and the onset of the growing season were well reproduced by the SPA model. However, difficulties with simulating the evergreen forest at Gludsted is evident with more variation modelled than given by the observations, and a lag of the start of the growing season when compared to the observations. The evergreen plant functional type in SPA lacks a labile / non-structural carbohydrate store needed to driver rapid leaf expansion with the onset of spring; instead leaf expansion is dependent on available photosynthate on a given time step. Therefore, SPA's LAI is lower early in the growing season resulting in a biased slow photosynthetic activity and an underestimate in the magnitude of NEE as seen at Gludsted (Fig. 4). Voulund alternates between winter and spring barley for the calibration period starting with winter barley in 2009. Note that the whole observed time series of NEE at Voulund is shown together with model NEE of both winter and spring barley (Fig. 4c and d). While the phenology and amplitude are well captured for spring barley at Voulund, SPA is not able to capture the seasonal amplitude of the winter barley that seems to be more sensitive to the meteorological drives, and seasons with harder winters had lower NEE peaks in summer (winter 2010-11, 2012-13). At the grassland site Skjern Enge, NEE is for winter, spring, and the first part of the summer reasonably modelled. The difficulties for late summer and autumn arise from the management practises at the site, where both grazing and grass cutting are conducted, limiting NEE (Herbst et al., 2013). Although grazing is included in the SPA model, it does not simulate the same reduction in NEE.

   Examining the performance of SPA at a higher temporal resolution (Table 3), the correlations are better for hourly values than daily for the landcover classification having problems with the phenology (evergreen forest and winter barley), because SPA is capable of reproducing the diurnal variability. For the remaining landcover classifications $R^2$ and RMSE are improved when going from hourly to monthly averages of NEE. Zooming in on shorter time windows, the timing of the diurnal cycle are in accordance with measured NEE (see e.g. Fig S9), but the amplitude is underestimated by SPA.

## 4 Results

The model system was run from 2008 to 2014, with the first three years regarded as a spin-up period. In the following sections the terrestrial and marine surface fluxes will be presented first, followed by measurements of atmospheric $CO_2$ from Risø campus tower that will be used to assess the performance of the DEHM-SPA model system, and evaluate local impacts from fjord systems on atmospheric $CO_2$ concentrations.

### 4.1 Surface fluxes

#### 4.1.1 Biospheric fluxes

As shown in Fig. 5, the SPA model simulates an east-west gradient in NEE for both January and July in 2011. Larger values of NEE are found in the western part of Denmark, while the islands and Eastern Jutland have lower biosphere fluxes. This gradient follows the distribution of the individual landcover classifications (Supplement Fig. S8), their phenology and productivity, but also reflects the urbanisation which is denser in the eastern part of the country. During January, total ecosystem respiration dominates NEE. Evergreen forests and grasslands are well represented in Western Jutland and even though these landcover classes have GPP, they are still dominated by total ecosystem respiration, but their total ecosystem respiration can be higher than the other landcover classifications because of the contribution from the autotrophic respiration that in SPA depends on GPP. During July, the productivity is at its highest for all landcover classes dominating total ecosystem respiration resulting in negative NEE (Fig. 6), and the gradient across the country is more likely a result of the urbanization.

Figure 6 shows the average monthly contribution from each landcover classification to the country wide NEE, which inherently follows their productivity but also reflects the area covered by each landcove type with highest peaks for winter wheat and grasslands during June. During winter, the spread amongst the landcover classifications are smaller, but still with numerically larger monthly fluxes for the landcover classifications with largest area. Integrating over all landcover classification, the Danish terrestrial land surfaces is a net source of $CO_2$ to the atmosphere in the months from October to April, with the highest monthly release of 1,063$\pm$154 GgC month$^{-1}$ in December. From May to September, the biosphere is a net sink with a maximum uptake in June of -4,982$\pm$385 GgC month$^{-1}$. The total annual surface exchange of $CO_2$ between the atmosphere and Danish biosphere is -7,337$\pm$1,468 GgC yr$^{-1}$, where winter wheat has the largest contribution with -2,342 $\pm$1,045 GgC yr$^{-1}$.

#### 4.1.2 Marine fluxes

The air-sea $CO_2$ exchange in the Danish inner waters experience large seasonal variations, while the variations in the North Sea are less pronounces as illustrated by Fig. 7. The minerlisation in winter increases the surface water pCO$_2$ in the Danish inner waters resulting in outgassing of $CO_2$ to the atmosphere, while uptake occurs during spring and summer months following the decrease in surface water pCO$_2$ due to biological activities.

The simulated annual air-sea $CO_2$ exchange in the 105,000 km$^2$ covered by the Danish EEZ amounts to -422 GgC yr$^{-1}$. However, this number masks large spatial differences and monthly numerical larger fluxes (Fig 6). While the North Sea area

contained within the EEZ continuously had monthly uptakes in the range -73 GgC mon$^{-1}$ to -191 GgC mon$^{-1}$ with an annual accumulation of 1,765 GgC yr$^{-1}$, the monthly fluxes from the near coastal Danish inner waters varied in the range -46 GgC mon$^{-1}$ to 540 GgC mon$^{-1}$ annually releasing 1,343 GgC yr$^{-1}$ to the atmosphere.

## 4.2 Atmospheric $CO_2$ concentrations

The time series of measured and simulated $CO_2$ show good agreements (Fig. 8) with $R^2 = 0.77$ and RMSE = 4.87 ppm for daily averaged time series demonstrating that the model is capable of capturing the synoptic scale variability. Also, good statistical measures are obtained for the hourly time series with $R^2 = 0.71$ and RMSE = 5.95 ppm, but the short-term variability was not always fully captured by the model. All in all the evaluation shows that the model can capture the overall variability of the atmospheric $CO_2$ concentrations and fluxes. Moreover, the higher resolution in both transport model and surface fluxes results

in a better model performance in simulating atmospheric $CO_2$ concentrations (Fig. S10).

    To investigate the origin of the $CO_2$ simulated at the Risø site, concentration rose plots of simulated atmospheric $CO_2$ have been made (Fig. 9). The concentration rose shows the wind direction and associated $CO_2$ concentrations. Division has been made between seasons, and day and night time values both showing distinct seasonal and diurnal patterns. The highest values of $CO_2$ are obtained during winter, where very little diurnal variation is seen. During summer the lowest values are obtained

in particular during daylight, when photosynthesis occurs.

    The individual contribution from fossil fuel emissions, marine and biospheric exchanges to the atmospheric $CO_2$ (see Supplement Fig. S11 - S13) indicate that the biosphere contributes most to the variations simulated at Risø (Fig. S12) – both seasonally and daily. Emissions of fossil fuel experience little diurnal variability, but seasonally with the greatest contribution during autumn and winter (Fig. S11). Highest values are seen originating from the sectors encapsulating the city of Roskilde

and the capital region. In all seasons, the simulated oceanic contribution is negative, i.e. indicating uptake of atmospheric $CO_2$, but the marine contribution is small with little variation (Fig. S13). The less negative values in autumn and winter may be a result of the simulated outgassing of $CO_2$ from the Baltic Sea and Danish inner waters during the winter season (Lansø et al., 2015), which however is still dominated by the uptake by global open oceans.

    The local impact from Roskilde Fjord is difficult to detect in the marine concentration plots. Flux measurements at Roskilde

Fjord have shown uptake of $CO_2$ during spring, while release in the remaining seasons (Mørk et al., 2016), which is accurately captured by the modelling system (Lansø et al., 2017). A footprint analysis of the Risø tower has shown that the fluxes from Roskilde Fjord have a contribution to the total $CO_2$ flux measured at the top of the 118 m high tower, but only minor, since fluxes over water typically are an order of magnitude smaller than fluxes over land (Sogachev and Dellwik, 2017). Therefore, we investigated a period with observed large outgassing from Roskilde Fjord - a storm event in October 2013 that was observed

to increase the monthly release of $CO_2$ in the fjord by 66 % (Mørk et al., 2016). The storm event passed Denmark on 28 October 2013, and at 06 UTC southerly winds transport air masses with higher $CO_2$ towards the Risø site (Fig. 10a), while at the same time a detectable increase in the oceanic contribution to the $CO_2$ concentration at the Roskilde Fjord system is seen (Fig. 10b). The model system simulates the small peak in the observed atmospheric $CO_2$ concentrations for 28 October (Fig. 11a) at the Risø site, but distinguishing between contributions from fossil fuel emissions, the biosphere and the ocean to the atmospheric

$CO_2$ concentration at Risø (Fig. 11b) reveals no oceanic impact, and hence no apparent influence from Roskilde Fjord during the storm event.

## 5 Discussion

### 5.1 Surface fluxes

The simulated annual uptake by deciduous forest of -284 $\pm$ 21 gC m$^{-2}$ yr$^{-1}$ for the period 2011-2014 is within the observed range of annual estimated NEE at Sorø from 1996 to 2009 spanning 32 gC m$^{-2}$ yr$^{-1}$ to -331 gC m$^{-2}$ yr$^{-1}$ (Pilegaard et al., 2011). Improvements to the evergreen plant functional type in SPA are needed, and an addition of a labile pool to the evergreen carbon assimilation would omit the seasonal lag (Williams et al., 2005). Such adjustments have already been made to the DALEC carbon assimilation system utilised by SPA (Smallman et al., 2017) substantially improving the representation of

terrestrial phenology, but not yet incorporated into SPA. The annual estimated uptake of -355 $\pm$ 41 gC m$^{-2}$ yr$^{-1}$ is in the low range of previous estimates of temperate evergreen forests with -402 gC m$^{-2}$ yr$^{-1}$ (Luyssaert et al., 2007) and Danish evergreen plantations of -503 gC m$^{-2}$ yr$^{-1}$ (Herbst et al., 2011). This could be caused by the slow leaf onset in spring, inhibiting the productivity at the beginning of the growing season.

Previous annual estimates at Danish agricultural field sites found carbon uptake of -31 gC m$^{-2}$ yr$^{-1}$ estimated from a mixed

agricultural landscape (Soegaard et al., 2003) and -245 gC m$^{-2}$ yr$^{-1}$ at a winter barley site (Herbst et al., 2011). The SPA-DEHM model system simulated annual uptakes for winter wheat of -252 $\pm$ 113 gC m$^{-2}$ yr$^{-1}$, and spring crops of -179 $\pm$ 28 gC m$^{-2}$ yr$^{-1}$, while winter barley had a smaller uptake of -82 $\pm$ 91 gC m$^{-2}$ yr$^{-1}$ with large standard deviation potentially resulting in small annual releases. The calibration and validation (Fig. 4c) shows difficulties in simulating the observed NEE during growing seasons for winter barley particularly after cold and snow covered winters. As pointed out in previous studies,

the crop modelling component in SPA could likewise be improved e.g. by inclusion of intra-seasonal crops (Smallman et al., 2014).

The current study estimated the Danish grasslands to be a sink of $CO_2$ with -210 $\pm$ 43 gC m$^{-2}$ yr$^{-1}$, which are similar albeit slightly smaller than the -267 gC m$^{-2}$ yr$^{-1}$ observed at the Skjern Enge grassland site during 2009-2011 (Herbst et al., 2013) and the -312 gC m$^{-2}$ yr$^{-1}$ observed at the Lille Valby grassland site, Denmark (Gilmanov et al., 2007). The European

grassland study by Gilmanov et al. (2007) found large variation in annual fluxes from grassland driven by environmental conditions and management practises at the sites varying from 171 gC m$^{-2}$ yr$^{-1}$ to -707 gC m$^{-2}$ yr$^{-1}$, but with most site having an annual uptake of carbon. As seen in Fig. 4e more work on grassland calibration could have been done, but the conditions and management regimes at Skjern Enge does not necessarily fit the rest of the Danish grasslands. With the chosen parameters, very comparable results were obtained indicating that such an additional calibration might not be advantageous.

A tilling approach has been used for the landcover classification in the SPA-DEHM modelling framework, including sub-grid heterogeneity in the model system. However, the seven landcover classes do not fully encompass the ecosystem variability in Denmark. Both grassland and agricultural other cover a broad range of sub-categories with both heather and meadow included in the grassland class, while agricultural other e.g. contains vegetables fields, hedgerows, woodland patches and uncultivated

land highlighting the need to adopt approaches allowing for generating novel spatially varying parameter sets (Bloom et al., 2016). Moreover, large urbanised areas are not accounted for in the current classes either. Adding more landcover classifications could give a better and more realistic surface description, if data for both calibration and validation for the lacking landcover classes, preferably from similar climatic region as Denmark, were available.

Compatible marine fluxes to previous estimates are obtained for the study region. On an annual basis, the Danish inner waters were found to be a source of 30 gC m$^{-2}$ yr$^{-1}$, which is agreeing with most previous studies. Wesslander et al. (2010) estimated Kattegat to act as a small sink of -14 gC m$^{-2}$ yr$^{-1}$ based on measurements of water chemistry, while Norman et al. (2013) on the contrary found a release of 19 gC m$^{-2}$ yr$^{-1}$ using a biogeochemical model of the Baltic Sea. Measurements from Danish fjords on the other hand consistently point towards these marine areas being annual sources of $CO_2$ with values in the range of 41 gC m$^{-2}$ yr$^{-1}$ to 104 gC m$^{-2}$ yr$^{-1}$ (Gazeau et al., 2005; Mørk et al., 2016). The current study estimates the North Sea to be a sink of -29 gC m$^{-2}$ yr$^{-1}$, which is very close to previous estimates, both measured and modelled, of -20 m$^{-2}$ yr$^{-1}$ and -25 m$^{-2}$ yr$^{-1}$ (Thomas et al., 2004; Prowe et al., 2009).

## 5.2 Atmospheric $CO_2$ and land-sea signals

WRF are in general capable of simulating the observed wind patterns, while the the overestimation of the wind velocity could lead to an overestimation of the atmospheric mixing. However, the SPA-DEHM modelling system resembles the synoptic and diurnal variability in the atmospheric $CO_2$ concentrations measured at Risø campus tower site. The variability at the Risø is dominated by the biospheric impact and fossil fuel emissions of $CO_2$. The signal from Roskilde Fjord is difficult to detect in the simulated $CO_2$ concentrations. Even when the marine contribution to the atmospheric concentration alone is examined, the Roskilde Fjord signal is hard to distinguish at the Risø campus tower. Moreover, sea breezes from the narrow Roskilde Fjord might be difficult to detect by the model system with its 5.6 km horizontal resolution.

As Roskilde Fjord previously by a footprint analysis was found to have an impact on the atmospheric $CO_2$ concentration at the top of the tower (Sogachev and Dellwik, 2017), a period with observations of large outgassing from Roskilde Fjord was examined to more clearly envisions its impact in the simulated concentration fields. Both the simulated and observed atmospheric $CO_2$ increased during the storm event on October 28 (Fig. 10a), but no concurrent increase was seen in the oceanic contribution to atmospheric $CO_2$ at the Risø site (Fig. 10b). This might be explained by the southerly winds that transported the $CO_2$ released from the fjord northward and away from the Risø campus tower, which is positioned in the southern part of the fjord. Moreover, in this study the increased flux from Roskilde Fjord was only caused by increased wind speed together with the impose diurnal cycle of marine $pCO_2$ (the diurnal amplitude for October was approximately 10 $\mu$atm), while measurements suggested that also an increase in surface water $pCO_2$ of approximately 300 $\mu$atm sustained the observed $CO_2$ flux (Mørk et al., 2016). The lack of such increase in surface water $pCO_2$ in the current modelling study, could explain why no impact on the simulated atmospheric $CO_2$ is seen from the marine component during the storm event. Thus, the results could indicate that *(i)* the narrow Roskilde Fjord was not sufficiently resolved in the current model framework, where the horizontal grid resolution is 5.6 km $\times$ 5.6 km, *(ii)* the surface water $pCO_2$ was not described in enough details in the model

system, *(iii)* Roskilde fjord is not in the footprint of the tower during the storm event or *(iv)* the fjord only has a minor impact on the atmospheric $CO_2$ concentrations at Risø.

However, the air-sea $CO_2$ exchange from the Danish inner waters (including all fjord, inner straits and Kattegat) has during winter an impact. Between November and February, the air-sea fluxes from the Danish inner water corresponds to 23 – 60 % of the monthly NEE (see Fig. 6). Moreover, the higher values of about 0.5 ppm in the concentration roses of the marine contribution to the atmospheric $CO_2$ concentrations at the Risø campus tower site in winter likewise emphasizes the marine impact, albeit the outgassing from the neighbouring Baltic Sea also have a contribution. Although the annual total numerical marine fluxes of 1,765 gC yr$^{-1}$ from the North Sea and 1,343 gC yr$^{-1}$ from the Danish inner waters are comparable to the sizes of annual NEE for individual landcover classifications (e.g. deciduous -987 gC yr$^{-1}$, evergreen -665 gC yr$^{-1}$ and grasslands -1467 gC yr$^{-1}$), the air-sea $CO_2$ fluxes are one order of magnitude smaller than the biospheric fluxes with 30 gC m$^{-2}$ yr$^{-1}$ for the Danish inner waters and -29 gC m$^{-2}$ yr$^{-1}$ for the North Sea and Skagerak.

## 5.3 Uncertainties in relation to surface exchanges of $CO_2$

Some of the largest uncertainties lie in the parameters underlying the terrestrial carbon cycle, in particular those governing allocation to plant tissues and their subsequent turnover. Most often these are based on maps of land cover or plant functional type, but parameter estimation via data assimilation analysis has shown substantial spatial variation of terrestrial ecosystem parameters within plant functional type groupings with consequences for carbon cycling predictions (Bloom et al., 2016). Increasing the quantity and type of observations available for data assimilation systems can have a significant impact on reducing uncertainty of model process parameters and simulated fluxes (Smallman et al., 2017). In particular, availability of repeated above ground biomass estimates was able to half the uncertainty of net biome productivity estimates for temperate forests (Smallman et al., 2017). Above ground biomass estimate are currently available from remote sensed sources (e.g. Thurner et al. (2014); Avitabile et al. (2016)) with future missions planned such as the ESA Biomass mission (Le Toan et al., 2011) and NASA GEDI (https://gedi.umd.edu/) providing high quality observations over the tropics and global scales respectively.

While SPA also uses DALEC to simulate carbon allocation and turnover, it is currently impractical to conduct a similar data assimilation analysis to optimise DALEC (or SPA) parameters based on comparison with observations of atmospheric $CO_2$ concentrations as this would require repetition of computationally intense simulations of atmospheric transport. While conducting such an analysis remains a future ambition we consider it to be out of scope for the current study, since the terrestrial surface fluxes in this study is constrained by one data stream consisting of EC measurements. This study has focused on surface fluxes over a relative short time period, and the model framework was capable of producing such fluxes, including their aggregated impact on atmospheric $CO_2$ concentrations (Fig. 4) with $R^2$ = 0.77 and RMSE = 4.87 ppm for daily values.

Uncertainties of the marine fluxes can be associated with both the choice of transfer velocity parameterisation, choice of wind speed product and the used surface water pCO$_2$ maps. Sensitivity analysis of global transfer velocity parameterisation based on $^{14}$C bomb inventories shows uncertainties of 20 %, while varying the applied wind speed products for these formulation increase the difference in the global annual flux by 40 % (Roobaert et al., 2018). Including empirical formulations of transfer velocity parameterisation in the analysis increased the sensitivity of wind speed product to nearly 70 %, while the uncertainty

of the parameterisation itself rose to more than 200 %. More than a doubling of the annual uptake by the usage of different transfer velocity formulations has likewise been shown for the study region (Lansø et al., 2015), while the choice of surface water $pCO_2$ map could change the study region from an annual sink to source of atmospheric $CO_2$ (Lansø et al., 2017). As shown by Roobaert et al. (2018) the ERA-Interim and the transfer velocity formulation by Ho et al. (2006) used in the present

study have a combined uncertainty estimate around 20 %. The improved data-driven near coastal Danish $pCO_2$ climatology better reflects the observed spatial dynamics and seasonality in the Danish inner waters (Mørk, 2015), albeit not diminishing the uncertainty related to surface maps of $pCO_2$, but reducing it.

## 6    Conclusions

By usage of the designed mesoscale modelling framework, it was possible to get a detailed insight into the spatio-temporal

variability of the Danish surface exchanges of $CO_2$ and the relative contribution from the different surface types. The simulated biospheric fluxes experienced an east-west gradient corresponding to the distribution of the landcover classes, their biological activity and the urbanization pattern across the country. The relative importance of the seven landcover classes varied throughout the course of the year. Grasslands had a high contribution to the monthly NEE through all seasons, while crop-lands influence grew from March to July. On an annual basis, winter wheat had the largest impact on the biospheric uptake

with -2,342 GgC $yr^{-1}$. However, the simulated biospheric uptake could benefit both from model improvement and divisions into more landcover classes. The marine fluxes, being subdivided into the North Sea including Skagerak and the Danish inner waters, had annual fluxes of opposite signs with the North Sea being a continuous sink of atmospheric $CO_2$ and the Danish inner waters experiencing small uptake in summer and release of $CO_2$ during winter resulting in a total marine annual uptake of -422 GgC $yr^{-1}$.

Good accordance between simulated and observed concentrations was found between modelled and observed atmospheric $CO_2$ concentrations for 2013 and 2014 at the Risø campus tall. The origin of the modelled $CO_2$ concentrations at Risø varied with biospheric fluxes having the largest impact on diurnal variability, while on a seasonal scale fossil fuel emissions also had a dominant role. The local impact from Roskilde Fjord was difficult to detect, while regional impact from the Baltic Sea and Danish inner Straits are apparent in winter. The results may indicate that Roskilde Fjord and its localised impact (i.e. at

the Risø campus tower site) on atmospheric $CO_2$ is not adequately resolved in the current model set-up or only have modest effect. Numerically, the annual fluxes from the North Sea and the Danish inner water were comparable in size to the annual net terrestrial fluxes from the individual landcover classifications.

In order to further examine the air-sea signal at the complex Risø site surrounded by a mosaic of fjord systems, land masses and the Danish inner water, more model experiments could be made, where a larger focus was put on other marine areas than

Roskilde Fjord as e.g. the Danish Inner Straits, Kattegat and the Baltic Sea. Although the total annual marine flux was small, it disguises large monthly variations, and further investigations could help to understand the carbon dynamics in coastal regions. A runoff component in the modelling system would moreover be beneficial for such studies.

*Code availability.* Scientist with an interset in the atmospheric chemical transport model, DEHM, can contact Jesper H. Christensen (jc@envs.au.dk) with enquiries. Scientist with an interest in the Soil-Plant-Atmosphere model, SPA, can visit its webpage (https://www.geos.ed.ac.uk/homes/mwilliam /spa.html) or contact Mathew Williams (mat.williams@ed.ac.uk).

*Competing interests.* The authors declare that there is no competing interests.

*Acknowledgements.* This study was carried out as part of a PhD study within the Danish ECOCLIM project funded by the Danish Strategic Research Council (Grant no. 10-093901). CarbonTracker CT2015 results provided by NOAA ESRL, Boulder, Colorado, USA from the website at http://carbontracker.noaa.gov have contributed to this work. Emissions inventories form IER, EDGAR and Aarhus University have likewise had an important contribution. This study has been supported by the TRY initiative on plant traits (http://www.try-db.org). The TRY initiative and database is hosted, developed and maintained by J. Kattge and G. Bönisch (Max Planck Institute for Biogeochemistry,

Jena, Germany). TRY is currently support by DIVERSITAS/Future Earth and the German Centre for Integrative Biodiversity Research (iDiv) Halle-Jena-Leipzig. Moreover, this work used eddy covariance data acquired and shared by the Danish HOBE Center for Hydrology - Hydrological Observatory fonded by the Villum Foundation and by the FLUXNET community. The FLUXNET eddy covariance data processing and harmonization was carried out by the European Fluxes Database Cluster, AmeriFlux Management Project, and Fluxdata project of FLUXNET, with the support of CDIAC and ICOS Ecosystem Thematic Center, and the OzFlux, ChinaFlux and AsiaFlux offices.

We are grateful for many fruitful discussions regarding the atmospheric measurements of $CO_2$ at the Risø tall tower and its applications in a modelling framework with Ebba Dellwik at DTU Wind energy.

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

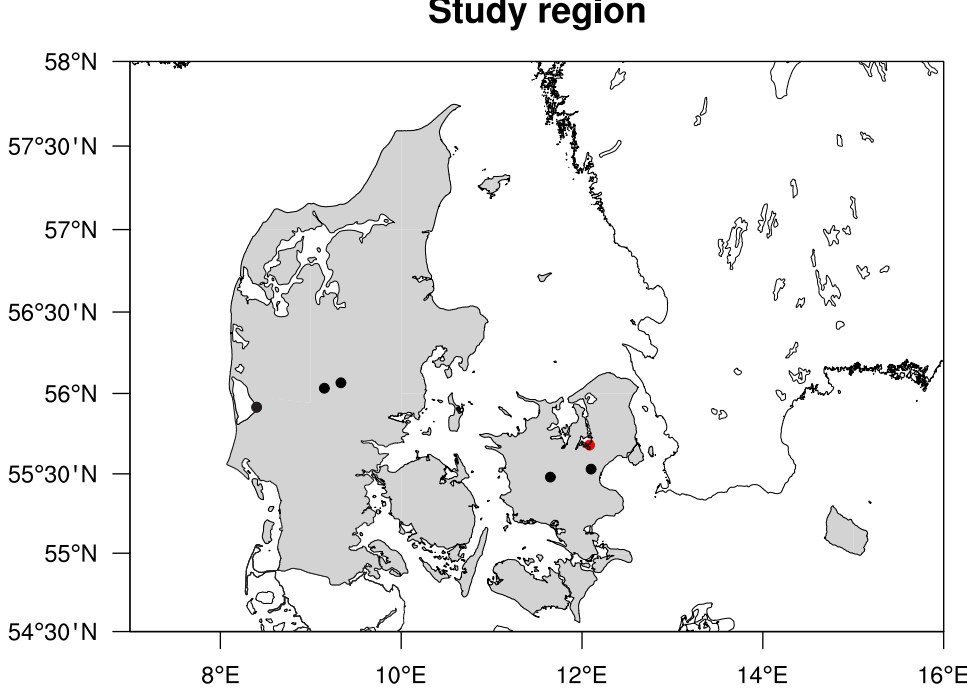

**Figure 1.** The study region of Denmark (land masses in grey) with the location of the five EC sites shown in black and the Risø camous tall tower site indicated in red.

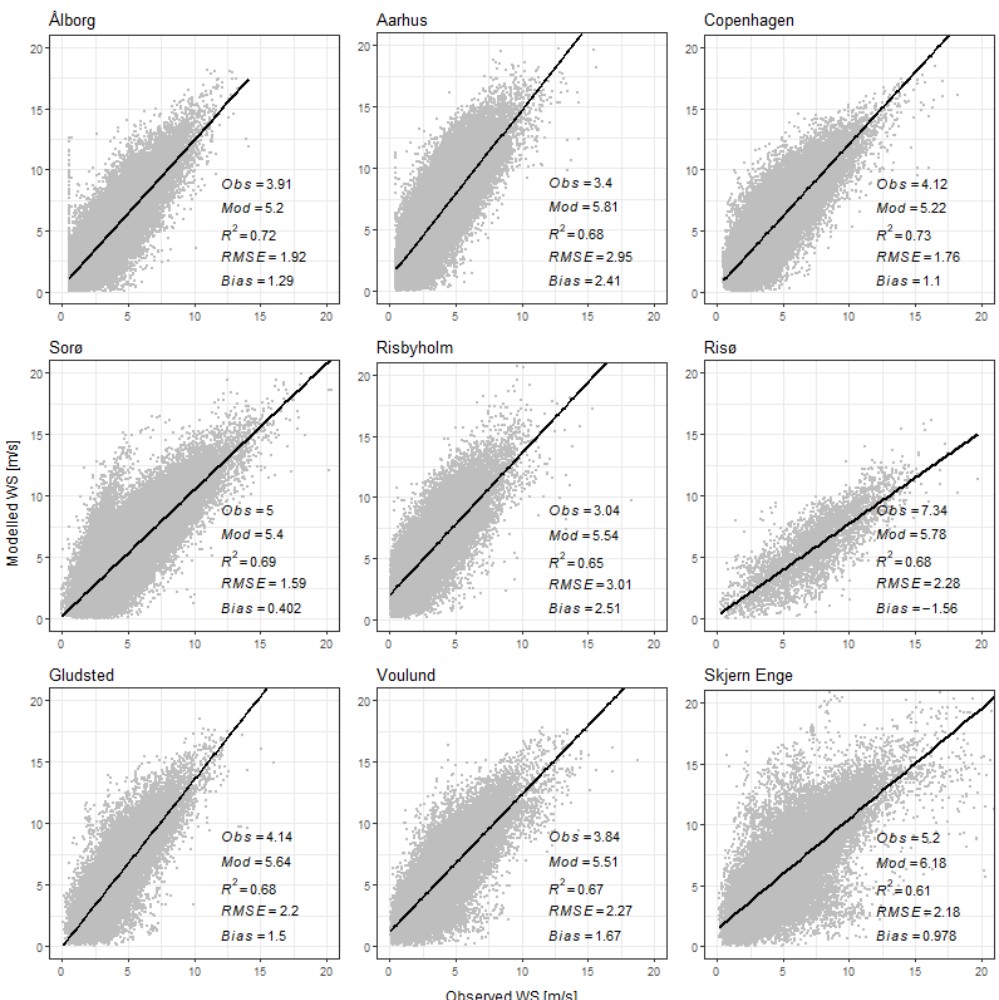

**Figure 2.** Scatter plots of measured versus modelled 10 m wind velocity for the nine sites used for evaluation of the meteorological drivers. Hourly average values are used for both simulated and measured wind velocities. Observed average wind velocity (Obs), simulated average wind velocity (Mod), correlation squared ($R^2$), root mean square error (RMSE) and bias are shown for each site.

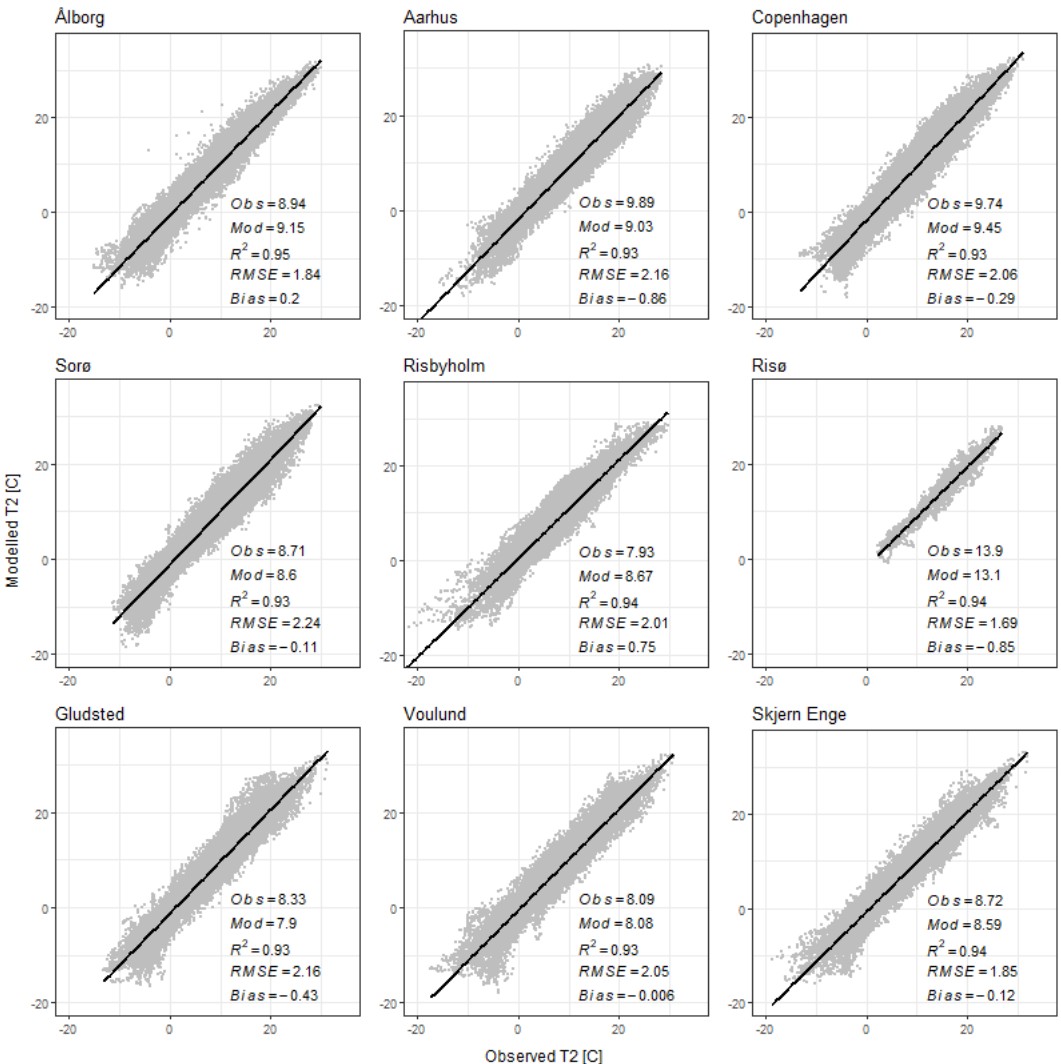

**Figure 3.** Scatter plots of measured versus modelled 2 m temperatures for the nine sites used for evaluation of the meteorological drivers. Hourly average values are used for both simulated and measured temperatures. Observed average 2 m temperature (obs), simulated 2 m temperature (mod), correlation squared ($R^2$), root mean square error (RMSE) and bias are shown for each site.

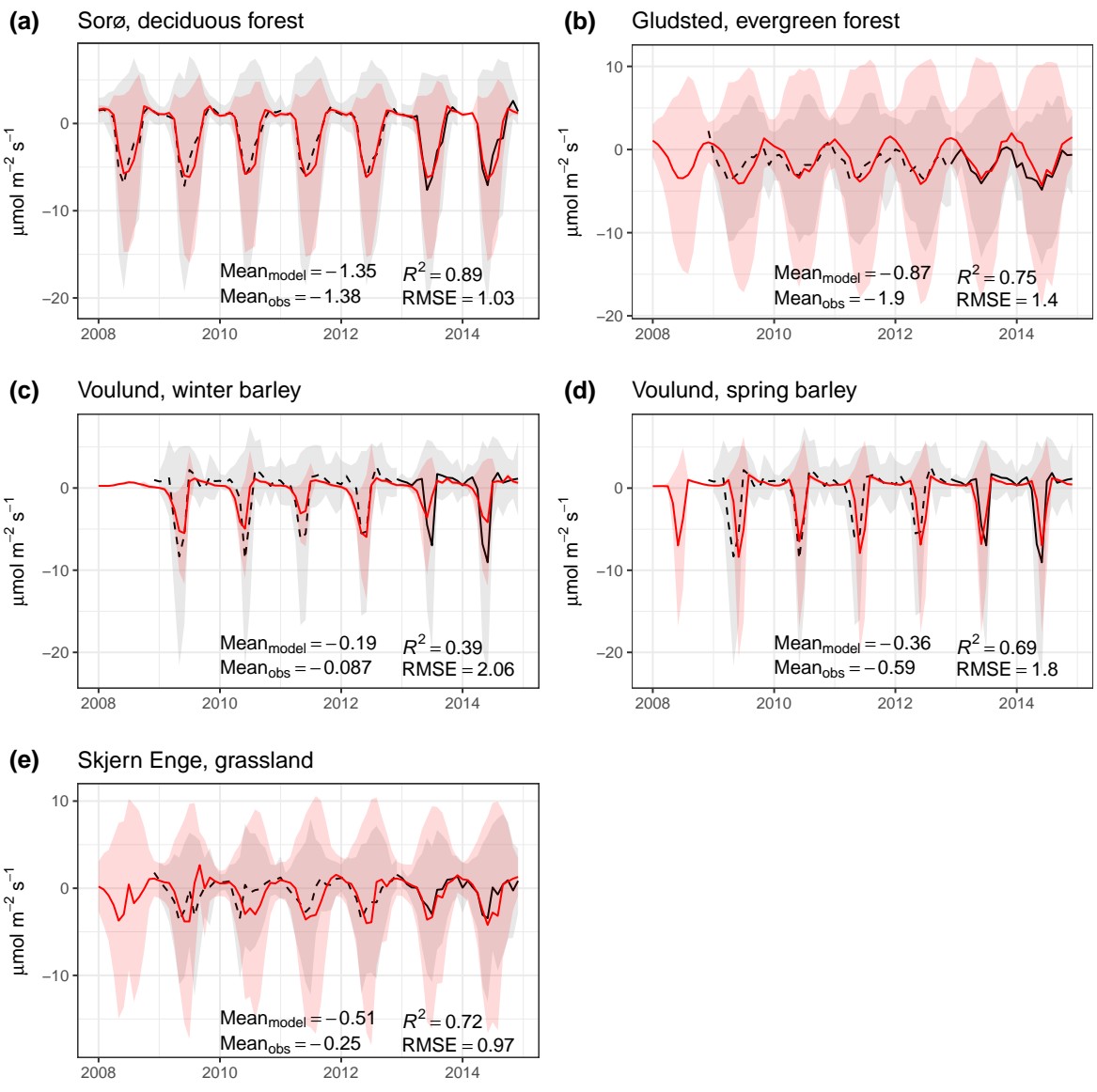

**Figure 4.** Monthly averaged values of measured (black dashed, calibration period; black solid, validation period) and simulated (red) net ecosystem exchange (NEE) for the Danish EC sites with measurements in the simulation period. The shaded areas show the standard deviations for the modelled and measured NEE calculated using hourly fluxes. The model mean ($\text{Mean}_{model}$), observational mean ($\text{Mean}_{obs}$), correlation squared ($R^2$) and root mean square error (RMSE) for the validation period (2013 and 2014) are shown for each site.

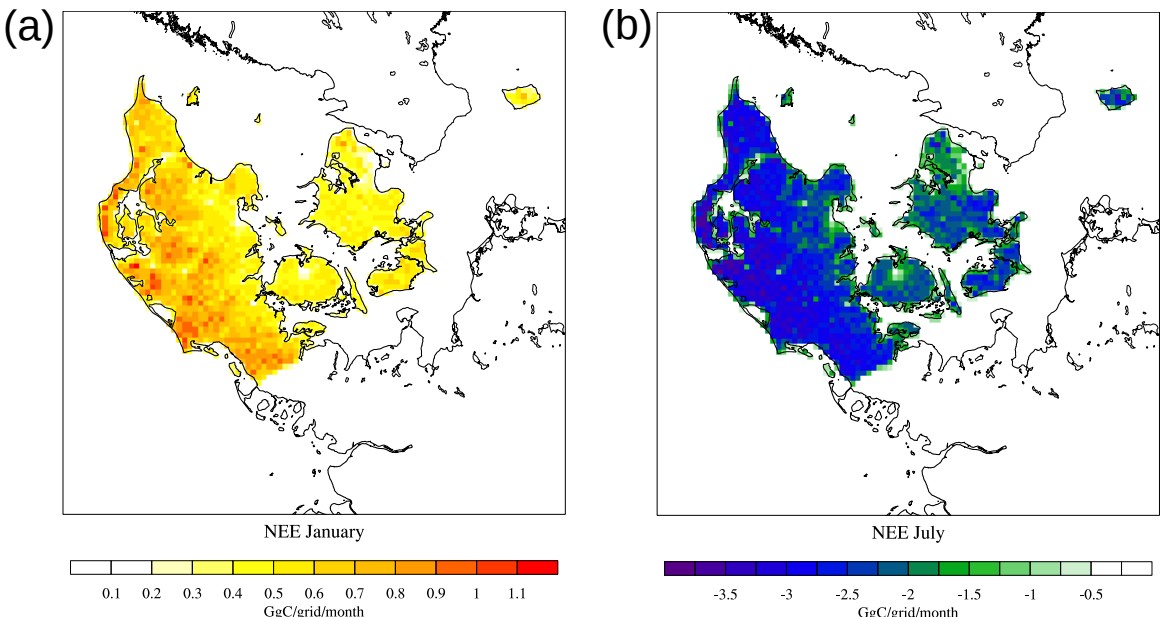

**Figure 5.** Net ecosystem exchange (NEE) for January **(a)** and July **(b)** 2011.

**Figure 6.** Total monthly average of NEE for each landcover classification for the simulation period of 2011-2014, together with the monthly air-sea $CO_2$ from the Danish marine areas that has been divided into the North Sea and Skagerak (NSSK) and Kattegat and the Danish straits (Inner waters).

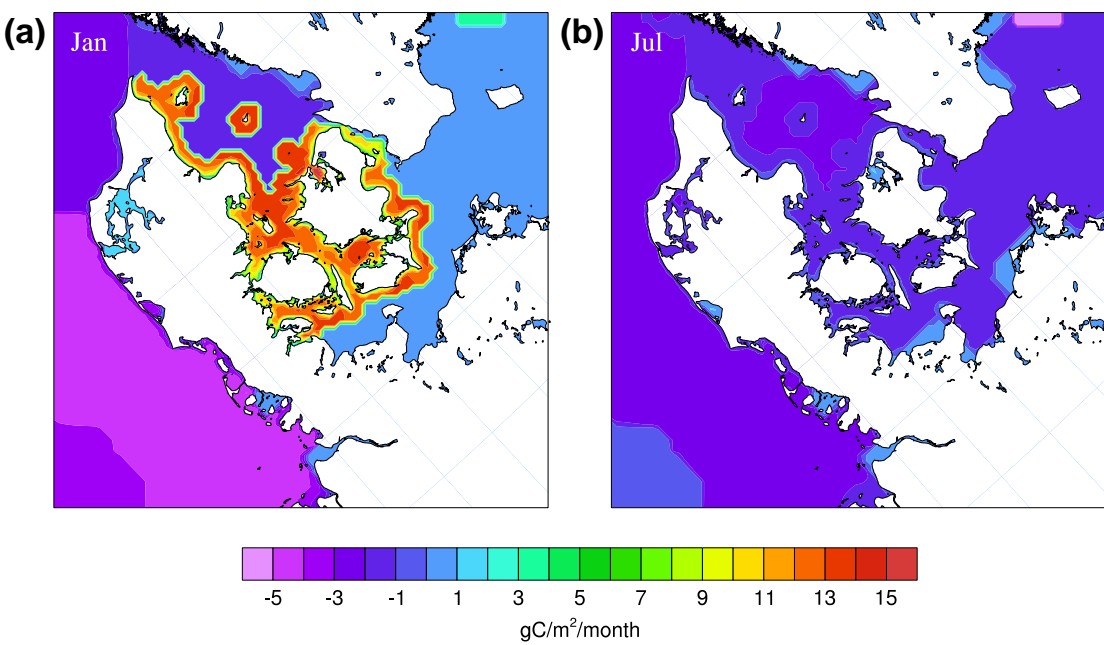

**Figure 7.** Simulated air-sea $CO_2$ exchange within the model framework for January **(a)** and July **(b)** 2011. The spatial resolution follows those of nest 4 from the DEHM model (i.e. 5.6 km x 5.6 km). The formulation by Ho et al., 2006 was used to calculate the air-sea CO2 exchange.

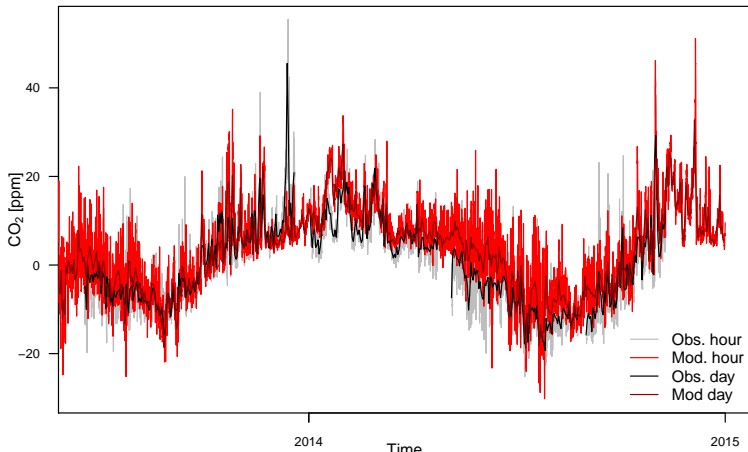

**Figure 8.** One-hour averages and daily averages of modelled and continuously measured atmospheric $CO_2$ at the Risø site for 2013-2014. The trends have been removed from the time series.

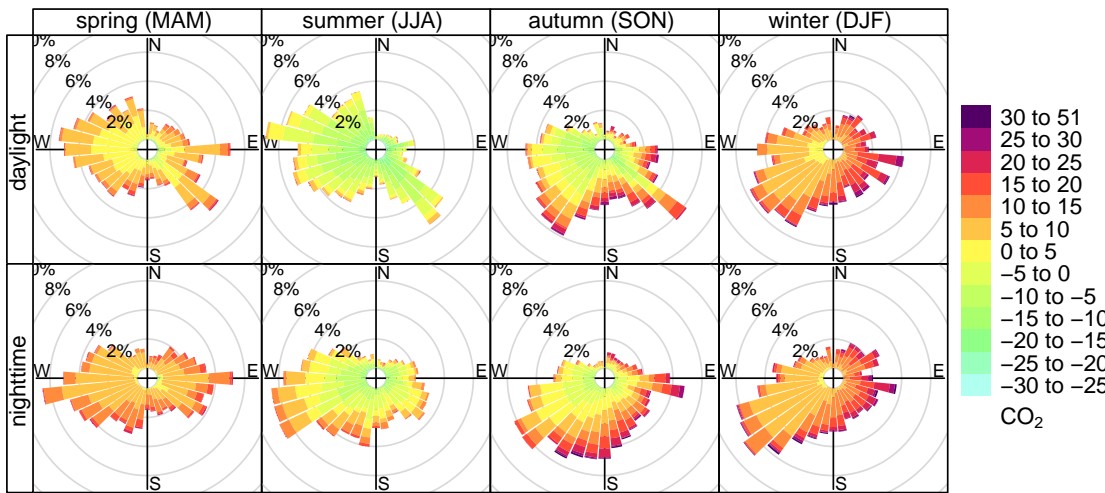

**Figure 9.** Concentration roses of modelled atmospheric $CO_2$ [ppm] at the Risø site for 2011-2014. The wind direction is split into 10° intervals and the frequency indicated by the concentric circles. The colours indicate the $CO_2$ concentrations with mean removed that have been transported to the site from the given wind directions.

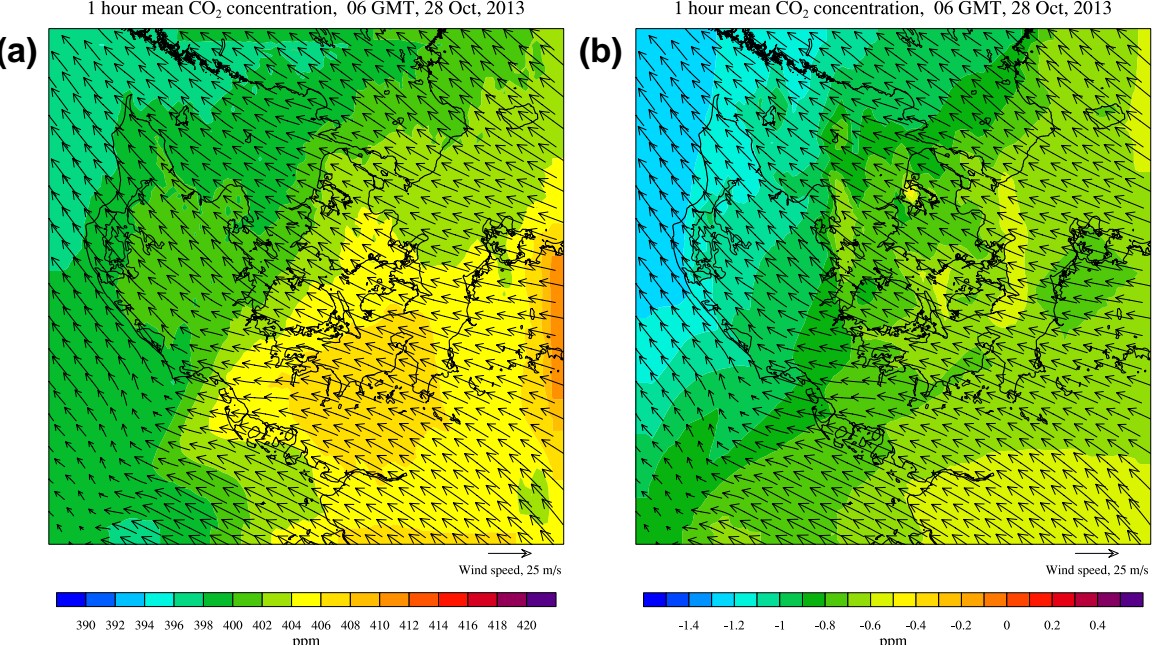

**Figure 10. (a)** Hourly averages of atmospheric $CO_2$ concentrations including the annual background across Denmark 28 October 2013 06 UTC during the October storm. **(b)** The contribution from the marine exchange alone to the hourly averaged atmospheric $CO_2$ concentration 28 October 2013 06 UTC. The less negative values at the Roskilde Fjord system indicates release of $CO_2$ to the atmosphere.

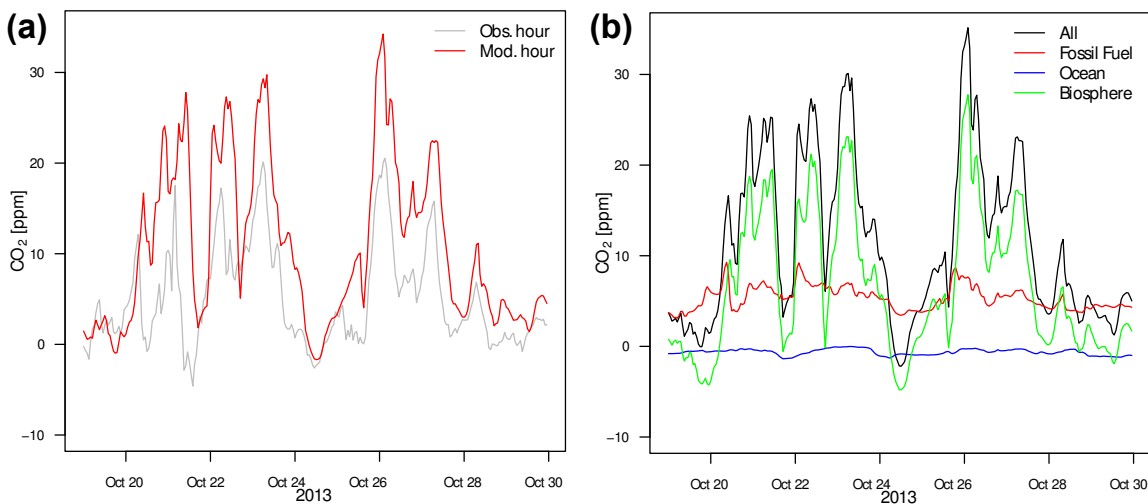

**Figure 11. (a)** One-hour averages of modelled and continuously measured atmospheric $CO_2$ at the Risø site for 19-29 October 2013 with annual means removed. **(b)** Contributions from fossil fuel emission, oceanic surface exchange and biospheric surface exchange to the atmospheric $CO_2$ concentration, shown as one-hour averages of modelled concentrations at the Risø site for the same period.

**Table 1.** Location of the sites used for evaluation of the meteorological drivers together with the time period from which measurements are used and the meteological variables included in the analysis. The measurements were obtained from Danish Hydrological observatory and exploratorium (HOBE), Fluxnet and Department of Environmental Science at Aarhus University (AU).

| Site | Location | Time period | Met par | Data source |
|------|----------|-------------|---------|-------------|
| Gludsted | 56°04′ N, 9°20′ E | 2009-2014 | T2,WS,WD,Rin | HOBE |
| Risbyholm | 55°32′ N, 12°06′ E | 2004-2008 | T2,WS,WD | Fluxnet |
| Skjern Enge | 55°55′ N, 8°24′ E | 2009-2014 | T2,WS,WD,Rin | HOBE |
| Sorø | 55°29′ N, 11°39′ E | 2006-2014 | T2,WS,WD,SRF | Fluxnet |
| Voulund | 56°02′ N, 9°09′ E | 2009-2014 | T2,WS,WD,Rin | HOBE |
| Risø Campus Tower | 55°42′ N, 12°05′ E | 2015 | T2,WS,WD | Fluxnet |
| Ålborg | 56°02′ N, 9°09′ E | 2004-2015 | T2,WS,WD | AU |
| Aarhus | 56°02′ N, 9°09′ E | 2004-2015 | T2,WS,WD | AU |
| Copenhagen | 56°02′ N, 9°09′ E | 2004-2015 | T2,WS,WD | AU |

**Table 2.** Location, species, landcover classification in the model system for the five Danish Eddy Covariance (EC) sites used for calibration and validation of the SPA model.

| Site | Location | Calibration | Validation | Species | LU in SPA-DEHM | Reference |
|------|----------|-------------|------------|---------|----------------|-----------|
| Gludsted | 56°04′ N, 9°20′ E | 2009-2012 | 2013-2014 | Norway Spruce | Evergreen forest | HOBE |
| Risbyholm | 55°32′ N, 12°06′ E | 2004-2008 | - | Winter Wheat | Winter wheat & winter crops | Fluxnet |
| Skjern Enge | 55°55′ N, 8°24′ E | 2009-2012 | 2013-2014 | Grass | Grassland | HOBE |
| Sorø | 55°29′ N, 11°39′ E | 2006-2012 | 2013-2014 | Beech | Deciduous forest | Fluxnet |
| Voulund | 56°02′ N, 9°09′ E | 2009-2012 | 2013-2014 | Spring & winter barley | Spring/winter barley | HOBE |

**Table 3.** Statistic metrics for the validation period (2013-2014) for the fives sites that have measurements of NEE during the validation period for hourly, daily and monthly values. Measured mean ($\text{mean}_{obs}$), modelled mean ($\text{mean}_{model}$), correlation squared ($R^2$), and root mean square error (RMSE) are shown for each site and temporal resolution.

| | $\text{mean}_{obs}$ | $\text{mean}_{model}$ | $R^2$ | RMSE | n |
|---|---|---|---|---|---|
| *Deciduous forest* | | | | | |
| Sorø hourly | -1.61 | -1.62 | 0.61 | 5.17 | 13746 |
| Sorø daily | -1.58 | -1.57 | 0.66 | 2.08 | 587 |
| Sorø monthly | -1.38 | -1.35 | 0.89 | 1.03 | 21 |
| *Evergreen Forest* | | | | | |
| Gludsted hourly | -1.95 | -0.88 | 0.59 | 6.71 | 17471 |
| Gludsted daily | -1.96 | -0.88 | 0.29 | 2.31 | 728 |
| Gludsted monthly | -1.95 | -0.87 | 0.75 | 1.41 | 24 |
| *Winter barley* | | | | | |
| Voulund hourly | -0.09 | -0.19 | 0.39 | 4.37 | 8759 |
| Voulund daily | -0.09 | -0.19 | 0.33 | 2.54 | 365 |
| Voulund monthly | -0.09 | -0.19 | 0.39 | 2.06 | 12 |
| *Spring barley* | | | | | |
| Voulund hourly | -0.59 | -0.36 | 0.47 | 5.00 | 8712 |
| Voulund daily | -0.59 | -0.36 | 0.55 | 2.56 | 363 |
| Voulund monthly | -0.59 | -0.36 | 0.69 | 1.95 | 12 |
| *Grasslands* | | | | | |
| Skjern Enge hourly | -0.26 | -0.52 | 0.40 | 6.18 | 17494 |
| Skjern Enge daily | -0.26 | -0.52 | 0.25 | 2.05 | 729 |
| Skjern Enge monthly | -0.25 | -0.51 | 0.72 | 0.97 | 24 |