# Peer review of "Simulating the atmospheric CO2 concentration across the heterogeneous landscape of Denmark using a coupled atmosphere-biosphere mesoscale model system"

_Biogeosciences, 2018_

## Referee Comment (RC1) · Anonymous Referee #1 · 8 Aug 2018

In this manuscript the authors use a coupled biosphere-atmosphere model SPA-DEHM to simulate the $CO_2$ surface exchanges and 3-D concentration fields over Denmark at a high horizontal resolution of 5.6 km. The simulated $CO_2$ surface fluxes and concentrations in the atmosphere are evaluated against observations from 5 EC flux tower and 1 tall tower atmospheric station in Denmark. Based on results from the simulation, annual $CO_2$ budget over this country is estimated and compared to others of similar latitudes and country size. A synoptic storm event is also investigated to examine contribution from land surface fluxes, ocean fluxes and fossil fuel fluxes, particularly impacts from the Roskilde Fjord system. The authors discuss sources of uncertainties regarding the simulation of $CO_2$ surface fluxes and propose future directions for model improvement. The manuscript is well structured and addresses variations of $CO_2$ surface fluxes and concentrations at different time scales. However, the authors seem to more focus on the biosphere model and overlook errors/uncertainties from atmosphere model that would influence simulation of $CO_2$ surface fluxes and concentrations; further, the scientific message of the manuscript is rather vague in its current form. I would like the authors to consider my questions and revise the manuscript before I recommend the publication of this paper. Detailed of my comments will be found in the following.

**General comments:**

1. This study uses a coupled biosphere-atmosphere model to simulate $CO_2$ surface fluxes and concentrations at a horizontal resolution of 5.6 km. Yet it's unclear whether and to what extent model performance will be improved with this fine resolution. Have you done any sensitivity test with coarser resolutions to show model improvement? Or have you compared results from your simulation to those from global models or regional models? How much are they different in terms of flux estimates and annual budgets?

2. The simulated $CO_2$ concentrations are evaluated against only one site in Denmark. It is relevant to include other European sites around the study area if there is any (e.g. MHD) to see: 1) whether the boundary conditions and regional transport are good enough for the nested coupled simulation; 2) whether the high resolution coupled model over Denmark improves representation of $CO_2$ variabilities at those sites.

3. The authors attribute uncertainties in simulating $CO_2$ surface fluxes using SPA to PFT-specific parameters regulating carbon allocation and turnover, as well as accuracy of PFT maps (especially agricultural-related landcover types). Have you examined whether the climate drivers and wind fields simulated by DEHM are in good quality? How much uncertainty in these variables?

4. In Section 4.3, the authors discuss the reasons why signals from the Roskilde Fjord system is not detected. While there are certainly representation errors in terms of grid size and uncertainties related to surface water $pCO_2$, another important source of uncertainties comes from transport errors. For example, the vertical resolution of DEHM is only 29 layers, which is rather coarse compared to its horizontal resolution. And the physical schemes related to boundary layer mixing are probably not capable to capture the land-sea breeze and diurnal variations of boundary layer height. This should be addressed and discussed in the manuscript.

5. For the "Abstract" section, it's too long and the description of the model setup is too detailed, which dilute the scientific message and significance that you would like to convey. An abstract should be concise, well-structured and focus on the most important findings and implication from the study, rather than simply listing the main results.

**Specific comments:**

Page 2 Lines 23–26 The statement is not accurate. There are numbers of studies on regional inversions over regions less covered by observational networks compared to US and Europe, like East Asia, South Asia, Amazonia, Siberia, etc., although with larger uncertainties.

Page 3 Lines 15–25: Please rephrase this paragraph. The description of the study area should be an independent section (see the next comment). And you should summarize here each section in the following manuscript.

Page 3 Line 26: There should be a section before model setup to describe the study area, including the landcover classification, coastal lines, major cities, important geographic characteristics (e.g., Roskilde Fjord system), etc.

Page 4 Line 5: How many vertical layers are there in the planetary boundary layers?

Page 5 Line 26: It would be better to mark the locations of EC flux sites and the tall tower for $CO_2$ measurements on a map.

Page 6 Line 4: How about the model performance on diurnal and daily variations of NEE? As you focused on a storm event during Oct. 19–29, 2013 in section 3.2, it would be better to have an idea of the capability of SPA to capture short-term variabilities.

Page 6 Line 10: Why rapid leaf growth in response to environmental drivers would cause a delay in spring photosynthesis?

Page 6 Line 25: What's included in "agricultural other"? It seems that it has substantial contribution to monthly GPP and respiration.

Page 6 Line 30: What is the altitude of this site? Can you further describe the dominant wind directions for each season (from observations), and potential influences from local pollution and vegetation activity? Again it would be better to have location of this station on the map.

Page 7 Line 3: How much does landcover classification vary over years? Do areas for certain landcover classifications vary a lot? If not, I would suggest to include the period 2012–2014 as well to calculate GPP, respiration, annual carbon budget, etc.

Page 7 Lines 14–24: It would be better to demonstrate the seasonal variations of GPP/respiration and contribution from landcover classifications with plots compared to tables.

Page 7 Line 16: The monthly contribution should also depend on the productivity of each land classification.

Page 8 Lines 14–17 Better to show the seasonal variations of $CO_2$ fluxes in coastal areas in a figure in the supplementary material.

Page 8 Line 31 Do you have observations of wind direction and speed corresponding to each

$CO_2$ measurements? It would be nice to plot concentration roses also based on observed CO2 and wind datasets, and see if model captures them well.

Page 11 Line 31: I think it's not precise to say that Roskilde fjord is not in the footprint of the tower. It could be in the footprint of the tower. As you mentioned, the marine signals cannot be seen because they are rather weak compared to land signals, or the current model is not capable to represent the complex topography, surface water pCO2 or transport. And as mentioned in the general comments, there are also uncertainties related to transport errors.

**Technical corrections:**

Throughout the manuscript, the authors use "land-use classifications" to indicate different vegetation types. In my opinion, it would be more appropriate to use "landcover classifications" as "land-use" emphasizes more human-induced influences (see https://oceanservice.noaa.gov/facts/lclu.html).

Page 2 Line 13: Data for which period?

Page 4 Line 11: "molefractions"

Page 6 Line 13: "by spring barley" -> "for spring barley"

Page 7 Line 3: "however,"-> "and"

Page 7 Line 10: "evident" -> "distributed" or "found"

Page 8 Line 6: "continuously" -> "persistent"

Page 8 Line 14: "hides" -> "masks"

Page 10 Line 30: "calibration of validation" -> "calibration and validation"

Page 10 Line 34: "appraises"?

Page 10 Line 34–35 For which period?

Figure 2,3,5,6 It would be better to: 1) mark locations of the flux towers and atmospheric station on the maps, as well as locations of major cities and Roskilde Fjord; 2) add lat/lon on all the maps for reference; 3) if possible, keep color bars and scales as same for comparisons between different panels and figures.

---

## Referee Comment (RC2) · Anonymous Referee #2 · 13 Sep 2018

This paper presents a coupled land-atmosphere model developed to simulate atmospheric CO2 mixing ratios over Denmark at high resolutions (here 5.6 km). The system includes a chemical transport model (DEHM) and a vegetation model (SPA) in addition to parameterized coastal fluxes and prescribed anthropogenic emissions of CO2. The evaluation of the SPA vegetation model was performed using eddy-covariance flux towers located across various ecosystems, while the simulated CO2 mixing ratios were compared to observations made at the Risø tower on the shore of Roskilde Fjord. The overall paper is well-written and the evaluation of the SPA model is well-documented.

Overall, the modeling system and more specifically the coupling between DEHM and SPA is well-described, with a tile approach similar to other systems (e.g. SiB-RAMS). The development of mesoscale systems for regional carbon studies is important to better understand the surface exchange of CO2 at finer scales. However, information related to the atmospheric model (here WRF), driving the DEHM chemical transport model, is completely absent. No information related to the WRF configuration is provided, and no evaluation of the model is included. This part of the system is critical, especially if this coupled system is intended to be used in regional inversions in the near-future. Even if not, the ability of the system to match the observed mixing ratios requires careful consideration of the meteorological performances. The paper has an extensive description and evaluation of SPA. The same should apply to the atmospheric model.

For this reason, the study requires a major revision with a dedicated section on the atmospheric model performances, and more specifically looking at the variables related to the transport of CO2 across the domain (i.e. wind speed and direction near the surface and above, and mixing heights). Because the Risø tower is located on the shore, an assessment of the land-sea air circulation should be a primary focus of this evaluation. The complexity of mesoscale circulation in cold waters and contrasted surface energy balance is challenging for meteorological models, and could impair the ability to simulate CO2 mixing ratios. With a description of the WRF simulations and an evaluation of the performances, this study would be suited for publication.

Another missing element for the SPA model evaluation is the absence of figures such as Taylor diagrams, or an illustration of the performances of SPA with different parameter values. A figure would help the reader to see the impact of your parameter optimization.

Additional comments are listed hereafter:

P1 – "Surface heterogeneity can be challenging to fully encompass by modelling

studies of CO2 surface exchanges, especially when it comes to land-sea boarders." Strange construction and confusing. Re-phrase. The first sentence should introduce the broader context of your study, and possibly the objectives in a broader context. Why do you need to understand the complexity of the land-sea border?

P1: "exploits" – Do you mean "explores"?

P1: "experienced" – showed, generated

P2: "These difficulties in simulating the local impact from the Roskilde Fjord might arise from" – What is the difference between (i) and (ii)? Isn't (ii) part of (i)? The third solution is not a difficulty but a possible physical reality. You should re-phrase the beginning of the sentence. And a fourth solution could be that the fluxes from the Fjord are small, hence not detectable.

P2 L10 – Eliminate is impossible. Any physical quantity has an associated uncertainty. What are these uncertainties? Cite studies that demonstrated our current uncertainties in the present carbon cycle are too large.

P2 L16 – Add references. Which surface observations are missing? All of them? Flux towers? Biomass measurements? Soil samples?

P2 – "These atmospheric inversions are capable of capturing the year to year changes in natural surface fluxes, the magnitude and distribution of regional fluxes, and distinguish between land and ocean fluxes (Le Quéré et al., 2015)." Several inter-comparison studies have shown large differences among inverse estimates. The study cited here is using aggregated inverse fluxes over large latitudinal bands or a land-ocean separation. This statement is very optimistic and very likely over-confident. See Peylin et al., (2013) for more details on global inversions.

P2: "atmospheric inversions are limited by the availability of atmospheric measurements" – And erroneous prior fluxes, errors in transport models, and simplified error covariances. Add citations related to limitations in inversions. Increasing the resolution

and denser networks do not fix all the problems encountered by global inversions.

P2 L32: "advantageous" – Do you mean "necessary" or "required"?

P3 L6-14: Most studies have ignored coastal fluxes because flux measurements and model estimates suggest that coastal fluxes are negligible compared to terrestrial fluxes in most areas. A brief comparison of existing studies to provide a range of coastal fluxes would be useful. While very large amount of carbon will be transported from the land to the deep ocean, the net surface fluxes remain small. You should justify better why you expect significant fluxes in your case.

P4: "such different things as" - delete.

P4 L6: "it's" – should be "its"

P4: You need to describe briefly the nesting. And DEHM is a chemical transport model using existing meteorological fields. These input fields come from a simulation or an existing product. In your case, if you used WRF simulations, you need to describe these as well, including WRF configuration and the domain setup.

P4: "towards the Southern Hemisphere"- Very confusing. What do you mean here? You have not coupled the full boundaries of your simulation domain? How are CO2 mole fractions coupled to your simulations?

P4 L19-21: This description is too succinct. You need to develop that part significantly. Describe the physical schemes used in WRF, the domain, and simulation period (re-initalizations or continuous run?)

Figure 1: Add the vegetation type next to the name of the site. How did you calculate the standard deviation? Is it the STD using 30-min fluxes? or from the parameter calibration? Provid more details on your shaded areas (STD's) for both model and data.

P6 For the Skjern Enge site, the uptake seems over-estimated by the model, as you

pointed out in the text. How much excess in uptake would that correspond to? You noted in the results, later in the paper, and in the abstract, how grassland plays a critical role in the annual uptake. Is it over-estimated by the SPA model?

P6 L27-28: Technically, the meteorological drivers come from the WRF model and not DEHM, to make it clear to the readers.

Section 2.3: This section is too succinct and provides very little details on the measurements. Which Picarro instrument was used? How often was it calibrated? Which standards did you use? Any publications looking at the data? Without a careful calibration, CRDS instruments from Picarro are not accurate enough to be used for CO2 studies. You need to document your measurements here.

P7 L9: This sentence is confusing. You can write instead "As shown in Figure 2, the SPA model simulates an east-west gradient…"

P7: "their grow patterns" – Unclear. Do you mean seasonal cycle? plant phenology?

P7: "population density" – Do you mean that the vegetation is less dense because of urbanization?

P7: "In winter, GPP is highest for evergreen, grassland and agricultural other." – Respiration is higher during that time of year. Why do you focus on GPP in winter? What about the net positive flux?

P7: "Respiration is less concentrated for individual land-use classes and the individual monthly contributions vary much less for respiration than GPP throughout the year" – How different are your parameter values for respiration across land use classes? Could you explain why? Is it a reasonable result?

P8: The Danish CO2 budget needs to be completed. When considering the total CO2 budget of a country, one needs to include the lateral fluxes (export/import) of agricultural production, and include all the sources of CO2 including animal livestocks. Otherwise you simply remove carbon from the country or from the food chain which

creates artificially a local sink in agricultural land not compensated by the emissions. If you want to discuss the national Danish CO2 budget, you need to consider all the components of the problem. I would suggest you simply remove this part, unless you want to develop it with the other exchanges of CO2.

P8: "Overall the model simulates the atmospheric CO2 quite well, indicating that the simulated surface exchange of CO2 is acceptable." Acceptable for what goal? How did you define the statistical success of your model? You need to discuss here what you want to accomplish with your system, and how you defined success.

P9: Autotrop[h]ic, heterotroph[h]ic

P10 L1-2: The range in values is so broad that the model can hardly fail. This is not very convincing to evaluate your simulated fluxes. Provide metrics instead (e.g. annual differences).

P10: "However, improvements to the evergreen plant functional type in SPA are needed" – Confusing. The model is fine (following the previous lines) but it needs improvement. Clarify why the model has to be improved.

P11: The discussion on the national CO2 budget is weak. As noted above, this part needs to be extended to the entire nation including all the components, as you noted in the discussion.

P11: The "land-sea signals" discussion seems to argue that fjord fluxes are still importat despite the limited impact on the modeled concentrations. If the tower location is a problem, you can sample your model in an optimal location to compute the maximum influence of the fjord on the CO2 mole fractions. You can look at the potential impact on the potential measurement locations. In any case, the fluxes are small. Is it really important at the annual scale? You need to provide numbers to demonstrate this statement. The section argues that fjords are important for the CO2 budget but without a clear demonstration.

Interactive
comment

P12: "to repeatedly simulate atmospheric transport to robustly quantify the impact of flux uncertainties on atmospheric CO2 concentrations due to their computational requirements." – Clarify. Why is SPA involved in atmospheric transport? What computational requirements?

P12: [at] a satisfactorily level

P12: "a satisfactorily level" – Again, you need to define why this is satisfying.

P12: "The usage of satellite retrievals by data assimilations systems and their accompanying improvements moreover highlights the future enhancement to the current modelling framework, where satellite products could be utilized for upscaling reducing the related error." – Very confusing sentence. Re-phrase. Which satellite data? What are the accompanying improvements? Future enhancements of what?

P12: "could be utilized for upscaling reducing the related error" – Which error?

P12: "while the choice of surface map could change the study region from an annual sink to source of atmospheric" – You need to clarify two things here. First, if you remove land from your map, you will make the fjord or the coast more important. What do you mean by "change the study region"? And second, even if you double your coastal flux, what would be the conclusions compared to the biosphere and the fossil fuel emissions? Globally, it matters, but regionally, aren't the conclusions unchanged?

Conclusions: Are there any measurements available to evaluate your coastal fluxes? Fig 2: Fonts are too small.

Fig 3: Your caption should include more information. Which driver data? at what resolution? and which formulation did you use?

Fig 4: "annual mean values" – Did you compute a running mean for each day of the year? or a trend?

Fig 6: Are these concentrations at the exact hour or hourly averages?

Fig A1: Fonts are too small. Caption needs additional information. Which model was used? At what resolution? . . .

Table 1: The references should list papers instead of data bases. Are there any papers documenting the flux data you used?

---

## Author Comment (AC1) · 31 Oct 2018

We would like to thank Reviewer 1 for taking his/hers time to complete the review of our paper and provide very constructive comment that will lead to an improvement of the quality of our manuscript. We appreciate that the reviewer acknowledges our well-structured manuscript. Reviewer 1 points out that our scientific results must be emphasized. This will be accomplished by strengthening the storyline of the manuscript with a deeper focus on coastal vs. land fluxes in the introduction and throughout the manuscript, while incorporating the suggestions made by both reviewers. As both Reviewer 1 and Reviewer 2 have asked for additional analysis and figures, we will add a supplement to the manuscript, where some of the new figures will be included.

The general and specific comments and suggestions made by Reviewer 1 will be addressed below, and outlines how the manuscript will be revised.

General comments: 1. This study uses a coupled biosphere-atmosphere model to simulate CO2 surface fluxes and concentrations at a horizontal resolution of 5.6 km. Yet it's unclear whether and to what ex-tent model performance will be improved with this fine resolution. Have you done any sensitivity test with coarser resolutions to show model improvement? Or have you compared results from your simulation to those from global models or regional models? How much are they different in terms of flux estimates and annual budgets?

Reply: To show how the results are improved with the higher spatial resolution we will add the following: Extract the annual CT optimised biosphere Danish flux and compare it the national fluxes obtained by the developed model system. The CT fluxes is 1 degree by 1 degree, while our system has a resolution of 5.6 km by 5.6 km. Moreover, a comparison of the atmospheric concentration at the Risø tall tower site between the main domain and the 3 nests will be included, thus the atmospheric CO2 concentration will be compared for grids with different spatial resolution (from 150 km to 5.6 km). In particular, during the growing season the biosphere will have an impact on the concentrations measured at the tall tower site (see Fig. 1 at the end of this document).

2. The simulated CO2 concentrations are evaluated against only one site in Denmark. It is relevant to include other European sites around the study area if there is any (e.g. MHD) to see: 1) whether the boundary conditions and regional transport are good enough for the nested coupled simulation; 2) whether the high resolution coupled model over Denmark improves representation of CO2 variabilities at those sites.

Reply: Simulated atmospheric CO2 concentrations by the model system have previously been validated for Northern Europe (see Lansø et al., 2015), where comparison

**BGD**
to MHD together with other Northern European atmospheric sites showed that the model system was both capable of capturing the boundary conditions and the regional transport. In the manuscript, we will add a new section on atmospheric validation, and include this reference in a sentence:" Moreover, long-range transport and boundary conditions of atmospheric CO2 concentrations have previously been shown to be well captured by the model system in and around the study area (Lansø et al., 2015)."

3. The authors attribute uncertainties in simulating CO2 surface fluxes using SPA to PFT-specific parameters regulating carbon allocation and turnover, as well as accuracy of PFT maps (especially agricultural-related landcover types). Have you examined whether the cli-mate drivers and wind fields simulated by DEHM are in good quality? How much uncertainty in these variables?

Reply: The meteorological fields used to drive the DEHM model are provide by the WRF model, which is a highly used model with a worldwide community of registered users (https://www.mmm.ucar.edu/weather-research-and-forecasting-model. DEHM and its meteorological drives have previously been used in well validated air pollution studies (see e.g. Im et al., 2018). We will include a new section on the atmospheric drivers and a validation of these in order to shed light on these questions. An example from the analysis can be seen in the Fig. 2, which shows that 2 m temperature is well captured with R2 of above 0.92 for 9 measurement sites.

4. In Section 4.3, the authors discuss the reasons why signals from the Roskilde Fjord system is not detected. While there are certainly representation errors in terms of grid size and uncertainties related to surface water pCO2, another important source of uncertainties comes from transport errors. For example, the vertical resolution of DEHM is only 29 layers, which is rather coarse compared to its horizontal resolution. And the physical schemes related to boundary layer mixing are probably not capable to capture the land-sea breeze and diurnal variations of boundary layer height. This should be addressed and discussed in the manuscript.

BGD
Reply: In DEHM approximately 10 of the 29 vertical layers are in the boundary layer. The diurnal variability is well captured, which can be seen in Fig. 1 in the end of the document, which will be included in the manuscript. We agree that air-sea breezes most likely are difficult to capture at this scale. We will elaborate on this more in the discussion.

5. For the "Abstract" section, it's too long and the description of the model setup is too detailed, which dilute the scientific message and significance that you would like to convey. An abstract should be concise, well-structured and focus on the most important findings and implication from the study, rather than simply listing the main results.

Reply: The abstract will be shortened and re-written to allow for an emphasize on the main scientific message.

Specific comments: Page 2 Lines 23–26 The statement is not accurate. There are numbers of studies on regional inversions over regions less covered by observational networks compared to US and Europe, like East Asia, South Asia, Amazonia, Siberia, etc., although with larger uncertainties.

Reply: The aim of this sentence was to show that the accuracy of inversions at a higher spatial scale is limited by available measurements. Therefore, only studies within areas of dense observational network was mentioned. As pointed out by Reviewer 1, the wished-for message does not come across this sentence. However, this paragraph will be deleted to allow for space in the introduction to improve the storyline in relation to the coastal vs. terrestrial fluxes.

Page 3 Lines 15–25: Please rephrase this paragraph. The description of the study area should be an independent section (see the next comment). And you should summarize here each section in the fol-lowing manuscript.

Reply: As suggested by the reviewer, we will include a section describing the study area. Thus, all characteristic of the study area will be omitted from this paragraph.
Instead it will be included in a designated section for the study region. We will conclude the introduction with a short paragraph detailing the rest of the content of the paper: "Sect. 2 is dedicated to describe the study region, which is followed by a detailed description and validation of the atmospheric and biospheric model components of the developed model system in Sect. 3 Sect. 4 contains results, while discussion and conclusion follow in Sect. 5 and 6."

Page 3 Line 26: There should be a section before model setup to describe the study area, including the landcover classification, coastal lines, major cities, important geographic characteristics (e.g., Roskilde Fjord system), etc.

Reply: We will include a section describing the study area including the abovementioned characteristics. Moreover, the description of the instrumentation at the Risø site will be included here in its own subsection: "The study area comprises of Denmark, a country that is characterised by a mainland (Jutland) and many smaller islands all containing varied land mosaic of urban, forest and agricultural areas. With more than 7,300 km of coastline encircling approximately 43,000 km2 of land, many land-sea borders are found throughout the country adding to the complexity. Denmark is positioned in the transition zone between the Baltic Sea and the North Sea. Bordering the Baltic Sea, the Danish inner waters are rich on nutrients and organic material (Kulinski and Pempkowiak, 2011). This fosters high biological activity in spring and summer lowering surface water pCO2 allow-ing for uptake of atmospheric CO2. In winter, mineralisation increases pCO2 (Wesslander et al., 2010), and out-gassing of CO2 to the atmosphere takes place. The North Sea is a persistent sink of atmospheric CO2, where a continental shelf-sea pump removes pCO2 from the surface water and transport it to the North Atlantic Ocean (Thomas et al., 2004). This study uses the Danish exclusive economic zone (EEZ) to estimate the Danish air-sea CO2 exchange, as the coastal state (in this case Denmark) has the right to explore, exploit and manage all resources found within its EEZ (United Nations Chapter XXI: Law of the Sea, 1984). The Danish EEZ is approximately 105,000 km2.

BGD
A tiling approach with the seven most common biospheric landcover classification were selected for the current study including deciduous forest, evergreen forest, winter wheat and other winter crops, winter barley, spring barley and other spring crops, grassland and agricultural other, but excluding urbanised areas. The agricultural other landcover classification includes all agricultural that does not classify as grain crop types (winter wheat, winter barley and spring crops), and as such find root crops, fruits, corn, hedgerows and agricultural 'undefined' are included in this classification. This classification cor-responds to the actual crop distribution of 2011 (Jepsen and Levin, 2013). Denmark is dominated by agriculture, and more than 60 % of the used classification is agricultural land."

Subsection on Observations of atmospheric CO2:" On the eastern inner shore of Roskilde Fjord tall tower atmospheric continuous measurements have been conducted at the Risø site between 2013 and 2015. The tower is located on small hill 6.5 m above sea level (Sogachev and Dellwick, 2017). Roskilde Fjord is a narrow microtidal estuary 40 km long with a surface area of 123 km\$^2\$ and a mean depth of 3 m (Mørk et al., 2016). The city of Roskilde with 50,000 inhabitants is positioned approximaetly 5 km southwest of the site, while Copenhagen lies 20 km east.

The tall tower continuous measurements of atmospheric CO2 concentrations at Risø were made by a Picarro G1301 placed in a heated building. The inlet was 118 m above the surface and the tube flow rate was 5 slpm. The Picarro was new and calibrated by the factory. The calibration was checked by a standard gas of 1000 ppm CO2 in atmospheric air (Air Liquide). During the measurement period from the middle of 2013 to the end of 2015, the instrument showed no other drift than the general in-crease in the global atmospheric concentration."

Page 4 Line 5: How many vertical layers are there in the planetary boundary layers? Reply: In DEHM approximately 10 of the 29 vertical layers are in the boundary layer. We will add this information to Page 4 line 5: "DEHM has 29 vertical levels distributed from the surface to the 100 hPa surface with approximately 10 levels in the boundary
layer."

Page 5 Line 26: It would be better to mark the locations of EC flux sites and the tall tower for CO2 measurements on a map.

Reply: A map is produced for the study area, where the locations of the EC flux sites and the tall tower are added (see Fig 3 at the of the document). The figure will be included in the section describing the study area.

Page 6 Line 4: How about the model performance on diurnal and daily variations of NEE? As you focused on a storm event during Oct. 19–29, 2013 in section 3.2, it would be better to have an idea of the capability of SPA to capture short-term variabilities.

Reply: A figure showing the diurnal variability of NEE will be included together with a short discussion of SPAs performance at the diurnal scale.

Page 6 Line 10: Why rapid leaf growth in response to environmental drivers would cause a delay in spring photosynthesis?

Reply: We acknowledge that the original sentence can cause some confusion, and thus to better explain, it will be changed from: "The evergreen plant functional type in SPA experiences phenological problems with rapid leaf growth in response to environmental drivers, which causes a delay in spring photosynthesis" To: "The evergreen plant functional type in SPA lacks a labile / non-structural carbohydrate store needed to driver rapid leaf expansion with the onset of spring; instead leaf expansion is dependent on available photosynthate on a given time step. Therefore, SPA's LAI is lower early in the growing season resulting in a biased slow photosynthetic activity and an underestimate in the magnitude of NEE as seen at Gludsted (Figure 1)."

Page 6 Line 25: What's included in "agricultural other"? It seems that it has substantial contribution to monthly GPP and respiration.

Reply: The agricultural other includes all other agricultural that does not classify as one of the grain crop types (winter wheat, winter barley and spring crops). Thus, here
we find root crops, fruits, corn, hedgerows and agricultural undefined. The information will be added to page 6 line 25: "The agricultural other landcover classification includes all agricultural that does not classify as grain crop types (winter wheat, winter barley and spring crops), and as such find root crops, fruits, corn, hedgerows and agricultural undefined are included in this classification"

Page 6 Line 30: What is the altitude of this site? Can you further describe the dominant wind directions for each season (from observations), and potential influences from local pollution and vegetation activity? Again, it would be better to have location of this station on the map.

Reply: The Risø tall tower is located 6.5 m above sea level just on the shore of Roskilde fjord. Unfortunately, we do only have measurement of wind speed and directions for six months, thus it is not possible to extract information on dominant wind direction as it can vary between season and year. But more details regarding local influence at the Risø site will be included in the section describing the study area.

Page 7 Line 3: How much does landcover classification vary over years? Do areas for certain land-cover classifications vary a lot? If not, I would suggest to include the period 2012–2014 as well to calculate GPP, respiration, annual carbon budget, etc.

Reply: The largest variations in landcover classification area are found for crops. The contribution from winter wheat, winter barley and spring crops to the total area covered in grain crops can for each vary with around 10% between years (Statistics Denmark, Statistikbanken). Since the variability only amounts to 10 % we will include the years 2012-2014 in the calculation of GPP and NEE. Therefore, the following line (page 7 line 3) will be deleted: "Given that the land-use classification corresponds to the actual distribution of 2011 an emphasis will be put on the terrestrial fluxes for this particular year during the analysis together with an estimation of the annual Danish carbon budget."

Page 7 Lines 14–24: It would be better to demonstrate the seasonal variations of GPP/respiration and contribution from landcover classifications with plots compared to

BGD
tables.

Reply: This suggestion will be implemented. See Fig. 4 at the end of this document as an example.

Page 7 Line 16: The monthly contribution should also depend on the productivity of each land classification.

Reply: For clarification the sentence will be changed form: "The monthly contributions to the country-wide total inherently reflect the total area for each land-use class." To "The monthly contributions to the country-wide total reflect both the productivity and total area for each landcover classification."

Page 8 Lines 14–17 Better to show the seasonal variations of CO2 fluxes in coastal areas in a figure in the supplementary material.

Reply: We would like to keep the figure of the seasonal CO2 fluxes in the coastal areas in the main part of the manuscript, as we will strengthen the storyline by focusing more on the coastal fluxes vs. the terrestrial fluxes.

Page 8 Line 31 Do you have observations of wind direction and speed corresponding to each CO2 measurements? It would be nice to plot concentration roses also based on observed CO2 and wind datasets, and see if model captures them well.

Reply: Sadly no, we do not have wind velocity and wind direction for the corresponding CO2 measurements. It would indeed have been interesting to plot the concentration roses for the measurements.

Page 11 Line 31: I think it's not precise to say that Roskilde fjord is not in the footprint of the tower. It could be in the footprint of the tower. As you mentioned, the marine signals cannot be seen because they are rather weak compared to land signals, or the current model is not capable to represent the complex topography, surface water pCO2 or transport. And as mentioned in the general comments, there are also uncertainties related to transport errors
Reply: We agree that the current formulation is not precise. It will be changed to: "... or (iii) Roskilde fjord is not in the simulated footprint of the tower, or (iv) the fjord only had a minor impact on the atmospheric CO2 concentrations at Risø."

Throughout the manuscript, the authors use "land-use classifications" to indicate different vegetation types. In my opinion, it would be more appropriate to use "landcover classifications" as "land-use" emphasizes more human-induced influences (see https://oceanservice.noaa.gov/facts/lclu.html). Reply: We apologize for this mash-up. The correction to landcover classifications will be made throughout the manuscript.

Technical corrections: We thank the reviewer for capturing these small errors. All the technical corrections will be conducted.

References

Im, U., Christensen, J. H., Geels, C., Hansen, K. M., Brandt, J., Solazzo, E., Alyuz, U., Balzarini, A., Baro, R., Bellasio, R., Bianconi, R., Bieser, J., Colette, A., Curci, G., Farrow, A., Flemming, J., Fraser, A., Jimenez-Guerrero, P., Kitwiroon, N., Liu, P., Nopmongcol, U., Palacios-Peña, L., Pirovano, G., Pozzoli, L., Prank, M., Rose, R., Sokhi, R., Tuccella, P., Unal, A., Vivanco, M. G., Yarwood, G., Hogrefe, C., and Galmarini, S.: Influence of anthropogenic emissions and boundary conditions on multi-model simulations of major air pollutants over Europe and North America in the framework of AQMEII3, Atmos. Chem. Phys., 18, 8929-8952, https://doi.org/10.5194/acp-18-8929-2018, 2018.

Lansø, A. S., Bendtsen, J., Christensen, J. 5 H., Sørensen, L. L., Chen, H., Meijer, H. A. J., and Geels, C.: Sensitivity of the air-sea CO2 exchange in the Baltic Sea and Danish inner waters to atmospheric short-term variability, Biogeosciences, 12, 2753–2772, https://doi.org/10.5194/bg-12-2753-2015, 2015.

Danmarks Statistik - Statistikbanken, https://www.dst.dk/da/Statistik/emner/erhvervslivets-sektorer/landbrug-gartneri-og-skovbrug/bedrifter

BGD
BGD
**Fig. 1.** Simulated atmospheric CO2 concentrations at the Risø tall tower site extracted from the main domain and three nests of DEHM. The rectifier effect is amplified in the main domain of DEHM.

---

## Author Response (AR1)

**Reply to Reviewer 1 for bg-2018-240**

We would like to thank Reviewer 1 for taking his/hers time to complete the review of our paper and provide very constructive comment that will lead to an improvement of the quality of our manuscript. We appreciate that the reviewer acknowledges our well-structured manuscript. Reviewer 1 points out that our scientific results must be emphasized. This will be accomplished by strengthening the storyline of the manuscript with a deeper focus on coastal vs. land fluxes in the introduction and throughout the manuscript, while incorporating the suggestions made by both reviewers. As both Reviewer 1 and Reviewer 2 have asked for additional analysis and figures, we will add a supplement to the manuscript, where some of the new figures will be included.

The general and specific comments and suggestions made by Reviewer 1 will be addressed below. The comments by reviewer is in normal font, while reply is in italic.

**General comments:**

1. This study uses a coupled biosphere-atmosphere model to simulate CO2 surface fluxes and concentrations at a horizontal resolution of 5.6 km. Yet it's unclear whether and to what extent model performance will be improved with this fine resolution. Have you done any sensitivity test with coarser resolutions to show model improvement? Or have you compared results from your simulation to those from global models or regional models? How much are they different in terms of flux estimates and annual budgets?

Sensitivity test have been made for the atmospheric CO2 concentration. Increasing the resolution of the model has improve its performance in simulating atmospheric CO2 concentrations as seen in Fig. S10. Moreover, it was necessary to increase the resolution to 5.6km x 5.6 km in order to include the best applicable surface water pCO2 maps for the Danish inner waters. This was essential for the study in investigating the impact of the marine vs. terrestrial fluxes.

2. The simulated CO2 concentrations are evaluated against only one site in Denmark. It is relevant to include other European sites around the study area if there is any (e.g. MHD) to see: 1) whether the boundary conditions and regional transport are good enough for the nested coupled simulation; 2) whether the high resolution coupled model over Denmark improves representation of CO2 variabilities at those sites.

Reply: Simulated atmospheric CO2 concentrations by the model system have previously been validated for Northern Europe (see Lansø et al., 2015), where comparison to MHD together with other Northern European atmospheric sites showed that the model system was both capable of capturing the boundary conditions and the regional transport. In the manuscript, we will add a new section on atmospheric validation, and include this reference in a sentence: "Moreover, long-range transport and boundary conditions of atmospheric CO2 concentrations have previously been shown to be captured by the model system across Northern Europe, in where the current study area is positioned, using observations from Mace head, Pallas, Westerland, the oil and gas platform F3, Lutjewad and Östergarnsholm (Lansø et al., 2015)."

3. The authors attribute uncertainties in simulating CO2 surface fluxes using SPA to PFT-specific parameters regulating carbon allocation and turnover, as well as accuracy of PFT maps (especially agricultural-related landcover types). Have you examined whether the climate drivers and wind fields simulated by DEHM are in good quality? How much uncertainty in these variables? *Reply: The meteorological fields used to drive the DEHM model are provide by the WRF model, which is a highly used model with a worldwide community of registered users (https://www.mmm.ucar.edu/weather-research-and-forecasting-model. DEHM and its meteorological drives have previously been used in well validated air pollution studies (see e.g. Im et al., 2018). We have included a new section on the atmospheric drivers (3.1.3 Meteorological drivers) and a section on the evaluation (3.1.4. Evaluation of Meteorological drivers) of these in order to shed light on these questions.*

4. In Section 4.3, the authors discuss the reasons why signals from the Roskilde Fjord system is not detected. While there are certainly representation errors in terms of grid size and uncertainties related to surface water pCO2, another important source of uncertainties comes from transport errors. For example, the vertical resolution of DEHM is only 29 layers, which is rather coarse compared to its horizontal resolution. And the physical schemes related to boundary layer mixing are probably not capable to capture the land-sea breeze and diurnal variations of boundary layer height. This should be addressed and discussed in the manuscript.

Reply: In DEHM approximately 10 of the 29 vertical layers are in the boundary layer. The diurnal variability is well captured, which e.g. can been seen in the original Fig. 5 of the manuscript, although the height of the night time boundary layer is too low. We agree that air-sea breezes from Roskilde fjord most likely are difficult to capture at this scale. The first paragraph in the discussion of the land-sea signals on page has been changed from:

"The SPA-DEHM modelling system resembles the synoptic and diurnal variability in the atmospheric CO2 concentrations measured at Risø site. The variability at the Risø site is dominated by the biospheric impact and fossil fuel emissions of CO2. The signal from Roskilde Fjord is difficult to detect in the simulated CO2 concentrations. Even when the marine contribution to the atmospheric concentration alone is examined, the Roskilde Fjord signal is hard to distinguish at the Risø tower. Rather the global and regional signals are obtained, e.g. the winter release of CO2 from the Baltic Sea resulting in less negative values for the concentration rose plot. Even though the oceanic contribution to the atmospheric short-term variability is low, oceanic impact is still important on monthly and annual scale time scales." To

"WRF are in general capable of simulating the observed wind patterns, while the the overestimation of the wind velocity could lead to an overestimation of the atmospheric mixing. However, the SPA-DEHM modelling system resembles the synoptic and diurnal variability in the atmospheric \chem{CO\_2} concentrations measured at Ris{\o} site. The variability at the Ris{\o} site is dominated by the biospheric impact and fossil fuel emissions of \chem{CO\_2}. The signal from Roskilde Fjord is difficult to detect in the simulated \chem{CO\_2} concentrations. Even when the marine contribution to the atmospheric concentration alone is examined, the Roskilde Fjord signal is hard to distinguish at the Ris{\o} tower. Moreover, sea breezes from the narrow Roskilde Fjord might be difficult to detect by the model system with its 5.6 km horizontal resolution."

5. For the "Abstract" section, it's too long and the description of the model setup is too detailed, which dilute the scientific message and significance that you would like to convey. An abstract should be concise, well-structured and focus on the most important findings and implication from the study, rather than simply listing the main results.

*Reply: The abstract has been shortened and re-written to allow for an emphasize on the main scientific message.*

**Specific comments:**

Page 2 Lines 23–26 The statement is not accurate. There are numbers of studies on regional inversions over regions less covered by observational networks compared to US and Europe, like East Asia, South Asia, Amazonia, Siberia, etc., although with larger uncertainties.

Reply: The aim of this sentence was to show that the accuracy of inversions at a higher spatial scale is limited by available measurements. Therefore, only studies within areas of dense observational network was mentioned. As pointed out by Reviewer 1, the wished-for message does not come across this sentence. However, this paragraph has been deleted to allow for space in the introduction to improve the storyline in relation to the coastal vs. terrestrial fluxes.

Page 3 Lines 15–25: Please rephrase this paragraph. The description of the study area should be an independent section (see the next comment). And you should summarize here each section in the following manuscript.

*Reply:* As suggested by the reviewer, we will include a section describing the study area. Thus, all characteristic of the study area will be omitted from this paragraph. Instead it will be included in a designated section for the study region. We have conclude the introduction with a short paragraph detailing the rest of the content of the paper:

"Sect. 2 is dedicated to describe the study region, which is followed by a detailed description and evaluation of the atmospheric and biospheric model components of the developed model system in Sect. 3 Sect. 4 contains results, while discussion and conclusion follow in Sect. 5 and 6."

Page 3 Line 26: There should be a section before model setup to describe the study area, including the landcover classification, coastal lines, major cities, important geographic characteristics (e.g., Roskilde Fjord system), etc.

*Reply:* We have included a section describing the study area including the abovementioned characteristics. Moreover, the description of the instrumentation at the Risø site will be included here in its own subsection:

"The study area comprises of Denmark, a country that is characterised by a mainland (Jutland) and many smaller islands, all containing varied land mosaic of urban, forest and agricultural areas. With more than 7,300 km of coastline encircling approximately 43,000 km2 of land, many land-sea borders are found throughout the country adding to the complexity.

Denmark is positioned in the transition zone between the Baltic Sea and the North Sea. Bordering the Baltic Sea, the Danish inner waters are rich on nutrients and organic material (Kulinski and Pemp-kowiak, 2011). This fosters high biological activity in spring and summer lowering surface water pCO2 allowing for uptake of atmospheric CO2. In winter, mineralisation increases pCO2 (Wesslander et al., 2010), and outgassing of CO2 to the atmosphere takes place.

The North Sea is a persistent sink of atmospheric CO2, where a continental shelf-sea pump removes pCO2 from the surface water and transport it to the North Atlantic Ocean (Thomas et al., 2004). This study uses the definition of the Danish exclusive economic zone (EEZ) to estimate the Danish air-sea CO2 exchange, as the coastal state (in this case Denmark) has the right to explore, exploit and manage all resources found within its EEZ (United Nations Chapter XXI: Law of the Sea, 1984). The Danish EEZ is approximately 105,000 km2.

A tiling approach with the seven most common biospheric landcover classification were selected for the current study including deciduous forest (3,348 km2), evergreen forest(1,870 km2), winter barley (1,211 km2), winter wheat and other winter crops (9,269 km2), spring barley and other spring crops( 5,368 km2), grassland (6,924 km2) and agricultural other (3,909 km2), but excluding urbanised areas. The agricultural other landcover classification includes all agricultural that does not classify as cereals and as such find root crops, fruits, corn, hedgerows and agricultural 'undefined' are included in this classification. This classification corresponds to the actual crop distribution of 2011 (Jepsen and Levin, 2013). Denmark is dominated by agriculture, and more than 60 % of the used classification is agricultural land."

Subsection on Observations of atmospheric CO2:

" One tall tower is found within the study area on the eastern inner shore of Roskilde Fjord. Here atmospheric continuous measurements have been conducted at the Risø campus site bewteen 2013 and 2015. The tower is located on small hill 6.5 m above sea level (Sogachev and Dellwick, 2017). Roskilde Fjord is a narrow microtidal estuary 40 km long with a surface area of 123 km2 and a mean depth of 3 m found in the sector 200- 360 relative to the Risø campus tower(Mørk et al., 2016). The city of Roskilde with aound 50,000 inhabitants is positioned approximately 5 km southwest of the site, while Copenhagen lies 20 km towards east.

The tall tower continuous measurements of atmospheric CO2 concentrations at Risø were carried out by the use of a Picarro G1301 placed in a heated building. The inlet was 118 m above the surface and the tube flowrate was 5 slpm. The Picarro was new and calibrated by the factory. The calibration was checked by a standard gas of 1000 ppm CO2 in atmospheric air (Air Liquide). During the measurement period from the middle of 2013 to the end of 2015, the instrument showed no other drift than the general increase in the global atmospheric concentration." Page 4 Line 5: How many vertical layers are there in the planetary boundary layers? *Reply: In DEHM approximately 10 of the 29 vertical layers are in the boundary layer. We will add this information to Page 4 line15:*

"DEHM has 29 vertical levels distributed from the surface to the 100 hPa surface with approximately 10 levels in the boundary layer"

Page 5 Line 26: It would be better to mark the locations of EC flux sites and the tall tower for CO2 measurements on a map.

Reply: A map is produced for the study area, where the locations of the EC flux sites and the tall tower are added (see Fig 3 at the of the document). The figure has been included in the section describing the study area.

Page 6 Line 4: How about the model performance on diurnal and daily variations of NEE? As you focused on a storm event during Oct. 19–29, 2013 in section 3.2, it would be better to have an idea of the capability of SPA to capture short-term variabilities.

Reply: In section "3.2.2 SPA evaluation" a table on the performance of SPA at hourly, daily and monthly scales has been included. Moreover, at representative figure for the performance at one specific site has been included in the Supplement. The following text has been added to line 29 p6:

"Examining the performance of SPA at a higher temporal resolution (Table 3), the correlations are better for hourly values than daily for the landcover classification having problems with the phenology (evergreen forest and winter barley), because SPA is capable of reproducing the diurnal variability. For the remaining landcover classifications the R2 and RMSE are improved when going from hourly to monthly averages of NEE. Zooming in on shorter time windows, the timing of the diurnal are in accordance with measured NEE (see e.g. Fig S9), but the amplitude is underestimated by SPA. "

Page 6 Line 10: Why rapid leaf growth in response to environmental drivers would cause a delay in spring photosynthesis?

*Reply:* We acknowledge that the original sentence can cause some confusion, and thus to better explain, it has been changed from:

"The evergreen plant functional type in SPA experiences phenological problems with rapid leaf growth in response to environmental drivers, which causes a delay in spring photosynthesis" To:

"The evergreen plant functional type in SPA lacks a labile / non-structural carbohydrate store needed to driver rapid leaf expansion with the onset of spring; instead leaf expansion is dependent on available photosynthate on a given time step. Therefore, SPA's LAI is lower early in the growing season resulting in a biased slow photosynthetic activity and an underestimate in the magnitude of NEE as seen at Gludsted (Figure 1)."

Page 6 Line 25: What's included in "agricultural other"? It seems that it has substantial contribution to monthly GPP and respiration.

Reply: The agricultural other includes all other agricultural that does not classify as one of the grain crop types (winter wheat, winter barley and spring crops). Thus, here we find root crops, fruits, corn, hedgerows and agricultural undefined. The information had been added to page6 line 25:

"The agricultural other landcover classification includes all agricultural that does not classify as grain crop types (winter wheat, winter barley and spring crops), and as such find root crops, fruits, corn, hedgerows and agricultural undefined are included in this classification"

Page 6 Line 30: What is the altitude of this site? Can you further describe the dominant wind directions for each season (from observations), and potential influences from local pollution and vegetation activ-ity? Again, it would be better to have location of this station on the map.

Reply: The Risø tall tower is located 6.5 m above sea level just on the shore of Roskilde fjord. Unfortunately, we do only have measurement of wind speed and directions for six months, thus it is not possible to extract information on dominant wind direction as it can vary between season and year. But more details regarding local influence at the Risø site has been included in the section describing the study area. Page 7 Line 3: How much does landcover classification vary over years? Do areas for certain landcover classifications vary a lot? If not, I would suggest to include the period 2012–2014 as well to calculate GPP, respiration, annual carbon budget, etc.

Reply: The largest variations in landcover classification area are found for crops. The contribution from winter wheat, winter barley and spring crops to the total area covered in grain crops can for each vary with around 10% between years (Statistics Denmark, Statistikbanken). Since the variability only amounts to 10 % we will include the years 2012-2014 in the calculation of GPP and NEE. Therefore, the following line (page 7 line 3) has been deleted: "Given that the land-use classification corresponds to the actual distribution of 2011 an emphasis will be put on the terrestrial fluxes for this particular year during the analysis together with an estimation of the annual Danish carbon budget."

Page 7 Lines 14–24: It would be better to demonstrate the seasonal variations of GPP/respiration and contribution from landcover classifications with plots compared to tables. *Reply: This suggestion has been implemented. See Fig. 4 at the end of this document as an example.*

Page 7 Line 16: The monthly contribution should also depend on the productivity of each land classification.

*Reply: For clarification the sentence has been changed form:*

"The monthly contributions to the country-wide total inherently reflect the total area for each landuse class."

То

"Figure 6 shows the average monthly contribution from each landcover classification to the country wide NEE, which inherently follows their productivity also reflects the area covered by each landcove type with highest peaks for winter wheat and grasslands during June"

Page 8 Lines 14–17 Better to show the seasonal variations of CO2 fluxes in coastal areas in a figure in the supplementary material.

*Reply:* We would like to keep the figure of the seasonal CO2 fluxes in the coastal areas in the main part of the manuscript, as we will strengthen the storyline by focussing more on the coastal fluxes v1s. the terrestrial fluxes.

Page 8 Line 31 Do you have observations of wind direction and speed corresponding to each CO2 measurements? It would be nice to plot concentration roses also based on observed CO2 and wind datasets, and see if model captures them well.

*Reply:* Sadly no, we do not have wind velocity and wind direction for the corresponding CO2 measurements. It would indeed have been interesting to plot the concentration roses for the measurements.

Page 11 Line 31: I think it's not precise to say that Roskilde fjord is not in the footprint of the tower. It could be in the footprint of the tower. As you mentioned, the marine signals cannot be seen because they are rather weak compared to land signals, or the current model is not capable to represent the complex topography, surface water pCO2 or transport. And as mentioned in the general comments, there are also uncertainties related to transport errors

*Reply: We agree that the current formulation is not precise. It will be changed to:*

"... or (iii) Roskilde fjord is not in the simulated footprint of the tower during the storm event, or (iv) the fjord only had a minor impact on the atmospheric CO2 concentrations at Risø."

Throughout the manuscript, the authors use "land-use classifications" to indicate different vegetation types. In my opinion, it would be more appropriate to use "landcover classifications" as "land-use" emphasizes more human-induced influences (see <a href="https://oceanservice.noaa.gov/facts/lclu.html">https://oceanservice.noaa.gov/facts/lclu.html</a>). Reply: We apologize for this mash-up. The correction to landcover classifications will be made throughout the manuscript.

**Technical corrections:**

We thank the reviewer for capturing these small errors. All the technical corrections will be conducted.

**References**

Im, U., Christensen, J. H., Geels, C., Hansen, K. M., Brandt, J., Solazzo, E., Alyuz, U., Balzarini, A., Baro, R., Bellasio, R., Bianconi, R., Bieser, J., Colette, A., Curci, G., Farrow, A., Flemming, J., Fraser, A., Jimenez-Guerrero, P., Kitwiroon, N., Liu, P., Nopmongcol, U., Palacios-Peña, L., Pirovano, G., Pozzoli, L., Prank, M., Rose, R., Sokhi, R., Tuccella, P., Unal, A., Vivanco, M. G., Yarwood, G., Hogrefe, C., and Galmarini, S.: Influence of anthropogenic emissions and boundary conditions on multi-model simulations of major air pollutants over Europe and North America in the framework of AQMEII3, Atmos. Chem. Phys., 18, 8929-8952, https://doi.org/10.5194/acp-18-8929-2018.

Lansø, A. S., Bendtsen, J., Christensen, J. 5 H., Sørensen, L. L., Chen, H., Meijer, H. A. J., and Geels, C.: Sensitivity of the air-sea CO2 exchange in the Baltic Sea and Danish inner waters to atmospheric short-term variability, Biogeosciences, 12, 2753–2772, https://doi.org/10.5194/bg-12-2753-2015, 2015.

Danmarks Statistik - Statistikbanken, https://www.dst.dk/da/Statistik/emner/erhvervslivets-sektorer/landbrug-gartneri-og-skovbrug/bedrifter

**Reply to Reviewer 2 for bg-2018-240.**

We would like to thank Reviewer 2 for taking his/hers time to complete the review of our paper and provide constructive comments that will lead to an improvement of the quality of our manuscript. We appreciate that the reviewer acknowledges the importance of studying the exchange of CO2 at finer scales to obtain a better understanding of the underlying processes. As pointed out be Reviewer 2, we have documented the coupling between SPA and DEHM well and provided a thorough evaluation of SPA, except for the inclusion of Taylor Diagrams, which we is now provided in a revised version of the manuscript. Reviewer 2 likewise asks for a validation of the meteorological drives used by the developed model framework including a description of the setting in our WRF configuration. DEHM driven by meteorological drivers from WRF has already been used in various well validated studies related to air pollution (see e.g. Im et al., 2018). But we agree that to provide a full analysis of the capability of our model system, we need to include this aspect in our manuscript. We have therefore include two new subsections; "3.1.3 Meteorological drivers" which includes a description of the specific setting of WRF used in this study, and "3.1.4 Evaluation of meteorological drivers" where an evaluation of the meteorological drives used in DEHM is conducted. As both Reviewer 1 and Reviewer 2 have asked for additional analysis and figures, we will add a supplement to the manuscript, where some of the new figures will be included.

**The specific comments and suggestions made by Reviewer 2 will be addressed below and moreover outlines how the manuscript has been revised. No minor comments (spelling, etc) are individually addressed, as they will all naturally be implemented.**

P1 – "Surface heterogeneity can be challenging to fully encompass by modelling studies of CO2 surface exchanges, especially when it comes to land-sea boarders." Strange construction and confusing. Re-phrase. The first sentence should introduce the broader context of your study, and possibly the objectives in a broader context. Why do you need to understand the complexity of the land-sea border?

*Reply: As per request from Reviewer 1, the abstract has been re-written and shortened. In doing so, the first sentences now introduces the boarder context of the study.*

P2: "These difficulties in simulating the local impact from the Roskilde Fjord might arise from" – What is the difference between (i) and (ii)? Isn't (ii) part of (i)? The third solution is not a difficulty but a possible physical reality. You should re-phrase the beginning of the sentence. And a fourth solution could be that the fluxes from the Fjord are small, hence not detectable.

Reply: No, (ii) is not intended to be a part of (i). (i) questions whether the fjord is resolved by the model grids, while (ii) questions whether the representation of surface water pCO2 is realistic and captures the large observed variability. Measurements have found that the surface water pCO2 in Roskilde Fjord can vary with 200 uatm (Mørk et al., 2016). To avoid confusion, the sentence is rephrase to:

"The inability to simulate the local impact from Roskilde Fjord might arise from; (i) the fjord not being adequately resolve by the model grids, (ii) the lack of a realistic representation of surface water pCO2, (iii) the fjord is not in the simulated footprint, and (iv) the fluxes from Roskilde Fjord are insignificant and thus not detectable".

P2 L10 – Eliminate is impossible. Any physical quantity has an associated uncertainty. What are these uncertainties? Cite studies that demonstrated our current uncertainties in the present carbon cycle are too large.

Reply: Agreed, this sentence was over optimistic. Eliminate is deleted and the paragraph is re-written including new references: "To have the best chance of accurately predicting the future evolution of the

carbon cycle, and its implications for our climate, it is important to minimise the uncertainties that exists presently (Carslaw et al., 2018). Enhanced knowledge and a better process understanding in ecological theory and modelling could potentially reduce the model structural uncertainties (Lovenduski and Bonan, 2017) which together with improvements in the spatial surface representation could minimise our current uncertainties".

P2 – "These atmospheric inversions are capable of capturing the year to year changes in natural surface fluxes, the magnitude and distribution of regional fluxes, and distinguish between land and ocean fluxes (Le Quéré et al., 2015)." Several inter-comparison studies have shown large differences among inverse estimates. The study cited here is using aggregated inverse fluxes over large latitudinal bands or a land-ocean separation. This statement is very optimistic and very likely over-confident. See Peylin et al., (2013) for more details on global inversions.

*Reply:* We agree that this sentence is optimistic, but the cited study was capable of distinguishing should fluxes, albeit at a very coarse resolution. However, this paragraph is deleted entirely to allow for space in the introduction to improve the storyline in relation to the coastal vs. land fluxes

P2: "atmospheric inversions are limited by the availability of atmospheric measurements"
And erroneous prior fluxes, errors in transport models, and simplified error covariances. Add citations related to limitations in inversions. Increasing the resolution and denser networks do not fix all the problems encountered by global inversions. *Reply: We agree that atmospheric inversions indeed are limited by more than atmospheric measurements. As mentioned above this paragraph is deleted.*

P3 L6-14: Most studies have ignored coastal fluxes because flux measurements and model estimates suggest that coastal fluxes are negligible compared to terrestrial fluxes in most areas. A brief comparison of existing studies to provide a range of coastal fluxes would be useful. While very large amount of carbon will be transported from the land to the deep ocean, the net surface fluxes remain small. You should justify better why you expect significant fluxes in your case.

Reply: We plan to improve the storyline of the manuscript by making this clearer in the introduction. The term coastal areas covers both coastal shelf seas and estuaries. In general, the air-sea CO2 exchange is per area numerical larger for estuaries than shelf seas (see Chen et al., 2013, Laruelle 2010 and Laruelle 2014), and can for estuaries vary in the range of -696 gC/m2/yr to 1,956 gC/m2/yr, while shelf seas have fluxes in the range of -153 gC/m2/yr to 180 gC/m2/yr (Chen et al., 2013). Denmark is bordered by the Baltic Sea and the North Sea, which are connected through the Danish straits and Kattegat. The Baltic Sea is a marginal sea that experiences large seasonal variability in their CO2 fluxes, with outgassing of CO2 in winter, while biologic activity allows for uptake during spring and summer, while the North Sea is a continental shelf sea with uptake throughout the year. Previous studies have estimated annual fluxes in the range of -34 to 20 gC/m2/yr for the Baltic Sea (Kuss et al., 2006, Norman et al., 2013, Wesslander et al., 2010), -40 to 19 gC/m2/yr for Kattegat (Gustafsson et al., 2014, Norman et al., 2013, Wesslander et al., 2010) and -17 gC/m2/yr for the North Sea. Laruelle et al., 2014 estimate the Baltic Sea to have a total annual uptake of 2.245 TgC/yr. Moreover, the few direct EC measurements in the Baltic Sea have found that upwelling events greatly increase the air-sea CO2 exchange (Kuss et al., 2006, Norman et al., 2013). Considering that the coastal sea area surrounding Denmark is almost thrice the size of the Danish land masses, the air-sea CO2 fluxes are thought to be of significance for the study region. We will elaborate in more details on this in the discussion, while also changing the paragraph between line 6 and line 14 in the introduction to:

"Heterogeneity can also be considerable in coastal oceans, and as with terrestrial surface fluxes, the high spatiotemporal variability leads to large uncertainties in estimates of coastal air-sea CO2 fluxes (Cai 2011, Laruelle et al., 2013). Coastal seas play an important role in the carbon cycle facilitating lateral transport of carbon from land to the open ocean, but almost 20 % of the carbon entering estuaries are released to the atmospheric, while 17 % of the carbon inputs in coastal shelfves comes from atmospheric exchange (Regnier et al., 2013).

The air-sea CO2 exchange is in general per area numerical larger for estuaries than shelf seas (Chen et al., 2013, Laruelle et al., 2014), and can annually for estuaries be as large as

1,958 gC/m2/yr while continental shelf seas have fluxes in the range of -154 gC/m2/yr to 180 gC/m2/yr (Chen et al., 2013). The large spatial and temporal heterogeneity of coastal ocean adds to the large uncertainty related to annual estimates of the air-sea CO2 exchange (Regnier et al., 2013). The observed high spatial and temporal variability (Kuss et al., 2006, Leinweber et al., 2009, Vandemark et al., 2011, Norman et al., 2013, Mørk et al., 2016) are not always included in marine models (Omstedt et al., 2009, Gypens et al., 2011, Kuznetsov et al., 2013, Gustafsson et al., 2015, Valsala et al., 2015], let alone taken into account in atmospheric mesoscale systems simulating CO2 (Sarrat et al., 2007, Geels et al., 2007, Law et al. 2008, Tolk et al., 2009, Broquet et al., 2011, Kretschmer et al., 2014). But a recent study found that short-term variability in the partial pressure of surface water CO2 (pCO2) substantially can affect simulated annual fluxes in certain coastal areas – in their case the Baltic Sea (Lansø et al., 2017). Moreover, direct eddy covariance (EC) measurements in the Baltic Sea have found that upwelling events greatly increase the air-sea CO2 exchange (Kuss et al., 2006, Norman et al., 2013)".

P4: You need to describe briefly the nesting. And DEHM is a chemical transport model using existing meteorological fields. These input fields come from a simulation or an existing product. In your case, if you used WRF simulations, you need to describe these as well, including WRF configuration and the domain setup.

*Reply: As already mentioned, a subsection including the WRF configuration will be added. The two-way nesting is included in section 3.1 and added to line 7 on page 4.*

"The two-way nesting replaces the concentrations in the coarser grids by the values from the finer grids."

P4: "towards the Southern Hemisphere"- Very confusing. What do you mean here? You have not coupled the full boundaries of your simulation domain? How are CO2 mole fractions coupled to your simulations?

Reply: *DEHM* only covers the Northern Hemisphere. Therefore, as boundary conditions, the model reads atmospheric mole fractions of CO2 vertically at the outer boundaries of the main domain of DEHM. All the outer boundaries of DEHM are facing the Southern Hemisphere. To avoid this confusion, the sentence is changed from

"Similarly, CT2015 three-hourly mole fractions of CO2 were used boundary conditions towards the Southern Hemisphere."

То

"Similarly, CT2015 three-hourly mole fractions of CO2 were read in as boundary conditions at the lateral boundaries of the main domain of DEHM."

P4 L19-21: This description is too succinct. You need to develop that part significantly. Describe the physical schemes used in WRF, the domain, and simulation period (reinitalizations or continuous run?).

Reply: A subsection, named "3.1.3 Meteorological drivers", containing this information, has been added to the manuscript: "The necessary meteorological parameters for DEHM were simulated by the Weather Research and Forecast Model (WRF) (Skamarock et al., 2008), nudged by six hourly ERA-Interim meteorology (Dee et al., 2011) continuously ran between 2008 and 2014, and was also used as initial and boundary conditions. In WRF the Noah Land Surface Model, Eta similarity surface layer and the Mellor-Yamada-Janjic boundary layer scheme were chosen to simulate surface and boundary layer dynamics. The CAM scheme was used for long and short-wave radiation, the WRF Single-Moment 5-class Microphysics scheme was applied for microphysical processes, and the Kain Fritsch scheme for cumulus parametrisation (Skamarock et al., 2008).

In WRF the same nests as in DEHM were chosen, and the meteorological outputs were saved every hour. To get the sub-hourly values that match the time step in DEHM, a temporal interpolation is conducted between the hourly time steps when DEHM is reading the hourly meteorological data. Furthermore, a correction of the horizontal wind speed is conducted in DEHM to ensure mass conservation and compliance with surface pressure (Bregman et al., 2003)." Figure 1: Add the vegetation type next to the name of the site. How did you calculate the standard deviation? Is it the STD using 30-min fluxes? or from the parameter calibration? Provide more details on your shaded areas (STD's) for both model and data.

*Reply:* For both model and observations the STD are calculated for hourly values. The information has been added to the figure caption.

P6 For the Skjern Enge site, the uptake seems over-estimated by the model, as you pointed out in the text. How much excess in uptake would that correspond to? You noted in the results, later in the paper, and in the abstract, how grassland plays a critical role in the annual uptake. Is it over-estimated by the SPA model?

Reply: At the Skjern Enge site, the model does indeed seem to overestimate the uptake when looking at the original Fig. 1. However, when examining the time period 2011-2014 the annual accumulated fluxes measured and simulated at this specific site are very similar with  $-232 \pm 102$  gC m2/yr for the measured values and  $-199 \pm 64$  gC m2/yr for the simulated fluxes. Of course, with larger differences between individual years. But overall, the model slightly underestimates the annual uptake at Skjern Enge by 14 %. The value reported by Herbst et al 2013 of -267 at Skjern Enge is for the period 2009-2011. Assuming all Danish grasslands were like Skjern Enge the model would have underestimated the total Danish annual uptake of grassland by about 200 GgC per year. Nevertheless, this is most likely not the case.

Section 2.3: This section is too succinct and provides very little details on the measurements. Which Picarro instrument was used? How often was it calibrated? Which standards did you use? Any publications looking at the data? Without a careful calibration, CRDS instruments from Picarro are not accurate enough to be used for CO2 studies. You need to document your measurements here. *Reply: The measurements from the Risø Tower have not previously been published. The required information has been added to section 2.3:*

"Tall tower continuous measurements of atmospheric CO2 concentrations at Risø were carried out by the use of a Picarro G1301 placed in a heated building. The inlet was 118 m above the surface and the tube flowrate was 5 slpm. At the onset of the measurements the Picarro was new and calibrated by the factory. The calibration was checked by a standard gas of 1000 ppm CO2 in atmospheric air (Air Liquide). During the measurement period from the middle of 2013 to the end of 2015, the instrument showed no other drift than the general increase in the global atmospheric concentration."

P7: "In winter, GPP is highest for evergreen, grassland and agricultural other." – Respiration is higher during that time of year. Why do you focus on GPP in winter? What about the net positive flux? *Reply: In the revised manuscript, a large focus has been put on NEE, and the above has been deleted.*

P7: "Respiration is less concentrated for individual land-use classes and the individual monthly contributions vary much less for respiration than GPP throughout the year" – How different are your parameter values for respiration across land use classes? Could you explain why? Is it a reasonable result?

Reply: In SPA a fraction of GPP is moved to a pool for autotrophic respiration. For all the landcover classes the turnover rate of this pool is 0.07. The fraction of GPP to autotrophic respiration varies between 0.32 to 0.55 amongst the landcover classification. The deciduous trees and the spring crops are more conservative with their carbon, and a smaller fraction of GPP is used for the autotrophic respiration than the evergreen and winter crops. Heterotrophic respiration is in SPA determined by the mineralisation rate, size of litter or soil organic matter pool, temperature and a temperature coefficient. Of parameters, only the mineralisation rate varies between the landcover classifications. In general, the crops have the highest mineralisation rates of litter and soil organic matter, reflecting that the residues from crops are easier degradable than residues from trees. Respiration occurs throughout the year. Heterotrophic respiration is controlled by temperature, thus if temperature increases, heterotrophic respiration will increase for all landcover classifications accordingly, and the mutual ratios might not be changed. Autotrophic respiration is directly dependent on the plant productivity: the more GPP, the more carbon can be put into the autotrophic respiration pool and the

larger amount of carbon can be respired. Since only a part of the total respiration is directly related to the GPP, less variation is seen for the monthly contributions in the original Table 3. As the focus in this section of the manuscript now mainly is on NEE, the above has been deleted.

P8: The Danish CO2 budget needs to be completed. When considering the total CO2 budget of a country, one needs to include the lateral fluxes (export/import) of agricultural production and include all the sources of CO2 including animal livestock. Otherwise you simply remove carbon from the country or from the food chain which creates artificially a local sink in agricultural land not compensated by the emissions. If you want to discuss the national Danish CO2 budget, you need to consider all the components of the problem. I would suggest you simply remove this part, unless you want to develop it with the other exchanges of CO2.

*Reply:* As it is currently not possible to include the remaining components for the national CO2 budget in the model framework, we will follow the recommendation of Reviewer 2 and delete this subsection in the manuscript.

P8: "Overall the model simulates the atmospheric CO2 quite well, indicating that the simulated surface exchange of CO2 is acceptable." Acceptable for what goal? How did you define the statistical success of your model? You need to discuss here what you want to accomplish with your system, and how you defined success.

Reply: With the constructed model framework we wish to accomplish a model system that is capable of simulating surface fluxes and atmospheric CO2 concentrations over Denmark at a high spatiotemporal resolution. One success criterion is to reproduce the temporal pattern at both diurnal and seasonal time scale when compared to measurements. The sentences on page 8 line 29 has been changed to: "All in all the evaluation shows that the model can capture the overall variability of the atmospheric CO2 concentrations and fluxes."

P10: "However, improvements to the evergreen plant functional type in SPA are needed" – Confusing. The model is fine (following the previous lines) but it needs improvement. Clarify why the model has to be improved.

Reply: Indeed, these sentences create some confusion. To clarify:

"Even though SPA experiences a lag in the seasonal onset for the evergreen forest, the annual estimated uptake of -386 gC m-2 yr-1 compares well with previous estimates of temperate evergreen forests with -402 gC m-2 yr-1 (Luysseart et al., 2007) and Danish evergreen plantations of -503 gC m-2 yr-1 (Herbst et al., 2011). However, improvements to the evergreen plant functional type in SPA are needed, and an addition of a labile pool to the evergreen carbon assimilation would omit the seasonal lag (Williams et al., 2005). Such adjustments have already been made to the DALEC carbon assimilation system utilised by SPA (Smallman et al., 2017), but not yet incorporated into SPA." Has become

"Improvements to the evergreen plant functional type in SPA are needed, and an addition of a labile pool to the evergreen carbon assimilation would omit the seasonal lag (Williams et al., 2005). Such adjustments have already been made to the DALEC carbon assimilation system utilised by SPA (Smallman et al., 2017), but not yet incorporated into SPA. The annual estimated uptake of -386 gC m-2  $^2$  yr-1 is in the low range of previous estimates of temperate evergreen forests with -402 gC m-2 yr-1 (Luysseart et al., 2007) and Danish evergreen plantations of -503 gC m-2 yr-1 (Herbst et al., 2011). This could be caused by the slow leaf onset in spring, inhibiting the productivity at the beginning of the growing season."

P11: The discussion on the national CO2 budget is weak. As noted above, this part needs to be extended to the entire nation including all the components, as you noted in the discussion. *Reply: As mentioned above, we will delete the section related to the national CO2 budget and consequently also the discussion section related to it.*

P11: The "land-sea signals" discussion seems to argue that fjord fluxes are still important despite the limited impact on the modelled concentrations. If the tower location is a problem, you can sample your model in an optimal location to compute the maximum influence of the fjord on the CO2 mole

fractions. You can look at the potential impact on the potential measurement locations. In any case, the fluxes are small. Is it really important at the annual scale? You need to provide numbers to demonstrate this statement. The section argues that fjords are important for the CO2 budget but without a clear demonstration.

Reply: A special focus has been put on Roskilde Fjord in the analysis because it is near the Risø campus tall tower site. Therefore, it is investigated whether a direct impact from the air-sea fluxes from Roskilde Fjord can be detected in the atmospheric CO2 concentration at the Risø site in the model system, which turns out to be difficult. We do not mean to state that the Danish fjords are of high importance. However, the air-sea CO2 exchange from all Danish marine areas (including all fjord, inner straits and Kattegat) has during winter an impact. Between November and February, the air-sea fluxes from the total Danish marine area corresponds to 20 - 47 % of the monthly NEE. As mentioned in the response to Reviewer 1, we plan to include the monthly air-sea CO2 fluxes from the Danish marine areas in the Table of NEE that moreover will be converted to a figure (see Fig. 1 at the end of this document). This aids in clarifying the section on the land-sea signal. Moreover, part of this section will be re-written to make sure this message gets across. The following has been added to the discussion line 33 p. 11:

"However, the air-sea CO2 exchange from the Danish inner waters (including all fjord, inner straits and Kattegat) has during winter an impact. Between November and February, the air-sea fluxes from the Danish inner water corresponds to 23 – 60 % of the monthly NEE (see Fig. \ref{fig:DK\_NEE}). Moreover, the higher values of about 0.5 ppm in the concentration roses of the marine contribution to the atmospheric CO2 concentrations at the Risø campus tower site in winter likewise emphasizes the marine impact give, albeit the outgassing from the neighbouring Baltic Sea also have a contribution. Although the annual total numerical marine fluxes of 1,765 gC yr-1 from the North Sea and 1,343 gC yr-1 from the Danish inner waters are comparable to the sizes of annual NEE for individual landcover classifications (e.g. deciduous -987 gC yr-1, evergreen -665 gC yr-1 and grasslands -1467 gC yr-1), the air-sea CO2 fluxes are one order of magnitude smaller than the biospheric fluxes with 30 gC m-2 yr-1 for the Danish inner waters and -29 gC m-2 yr-1 for the North Sea and Skagerak. "

P12: "to repeatedly simulate atmospheric transport to robustly quantify the impact of flux uncertainties on atmospheric CO2 concentrations due to their computational requirements." – Clarify. Why is SPA involved in atmospheric transport? What computational requirements? *Reply: What was meant was that repeating the simulation to determine the impact on atmospheric concentrations due to changes in surface CO2 exchange alone is out of scope for the current study The existing sentence:*

"While SPA also uses DALEC to simulate carbon allocation and turnover, it is currently impractical to conduct a similar data assimilation analysis or repeatedly simulate atmospheric transport to robustly quantify the impact of flux uncertainties on atmospheric CO 2 concentrations due to their computational requirements."

Has become

"While SPA also uses DALEC to simulate carbon allocation and turnover, it is currently impractical to conduct a similar data assimilation analysis to optimise DALEC (or SPA) parameters based on comparison with observations of atmospheric CO2 concentrations as this would require repetition of computationally intense simulations of atmospheric transport."

P12: "The usage of satellite retrievals by data assimilations systems and their accompanying improvements moreover highlights the future enhancement to the current modelling framework, where satellite products could be utilized for upscaling reducing the related error." – Very confusing sentence. Re-phrase. Which satellite data? What are the accompanying improvements? Future enhancements of what?

Reply: We agree that this sentence is poorly placed and out of scope for the paper. Thus, it has been removed it, and instead a sentence from earlier in the same paragraph has been revisited: "Increasing the amount of observational data used in data assimilation system have been found to reduce uncertainty in retrieved parameters and thus simulated carbon stocks and fluxes of CO 2

(Smallman et al., 2017); including all observations counting both in situ and satellite, Smallman et al. (2017) halved the uncertainty of the net biome productivity" To

"Increasing the quantity and type of observations available for data assimilation systems can have a significant impact on reducing uncertainty of model process parameters and simulated fluxes (Smallman et al., 2017). In particular, availability of repeated above ground biomass estimates was able to half the uncertainty of net biome productivity estimates for temperate forests (Smallman et al., 2017). Above ground biomass estimate are currently available from remote sensed sources (e.g. Thurner et al., 2014; Avitabilie et al., 2016) with future missions planned such as the ESA Biomass mission (LeToan et al., 2011) and NASA GEDI (https://gedi.umd.edu/)."

P12: "could be utilized for upscaling reducing the related error" – Which error? *Reply: See previous reply.*

P12: "while the choice of surface map could change the study region from an annual sink to source of atmospheric" – You need to clarify two things here. First, if you remove land from your map, you will make the fjord or the coast more important. What do you mean by "change the study region"? And second, even if you double your coastal flux, what would be the conclusions compared to the biosphere and the fossil fuel emissions? Globally, it matters, but regionally, aren't the conclusions unchanged? Conclusions: Are there any measurements available to evaluate your coastal fluxes? *Reply: The choice of surface map here refers to the choice of pCO2 map applied to the coastal region, while the study region refers to the Danish waters. The air-sea CO2 exchange is evidently sensitive to the surface water concentrations of CO2. If the product providing surface water CO2 is changed, the annual air-sea flux of CO2 will be altered and can even change sign. To avoid further confusion this has been clarified in line 24 p12 changing it from:*

"while the choice of surface map could change the study region from an annual sink to source of atmospheric CO2 (Lanso et al., 2017)."

to

"while the choice of surface water pCO2 map could change the study region from an annual sink to source of atmospheric CO2 (Lanso et al., 2017)."

Moreover, the total Danish coastal fluxes have been added to the figures showing monthly NEE for the different landcover classifications to show that on a monthly basis these coastal fluxes can be comparable to monthly fluxes of individual landcover classifications (see Fig. 1 at the end of this document). During the cause of the year the coastal fluxes for the study region, however, almost averages out to zero. Thus, if we double the monthly air-sea CO2 fluxes for the study region we would reach the same conclusion for the annual flux, because we have a coastal system that seasonally can shift between a source and a sink of CO2

Only few direct measurements of the air-sea CO2 exchange are available for the Baltic Sea and only for limited time periods (Roskilde Fjord (2012-2013), Arkona Sea 2002-2003, and short periods at Östergarnsholm). Only Roskilde Fjord is positioned with the study area of the current study. The applied monthly pCO2 maps has previously been compared to pCO2 measurements in Danish waters and were found to capture the seasonal cycle (Lansø 2016).

Fig 3: Your caption should include more information. Which driver data? at what resolution? and which formulation did you use?

Reply: The air-sea CO2 fluxes are calculated within the model framework at each time step, thus meteorological drives from WRF are used for these calculations. The spatial resolution follows those from the DEHM nest, and thus over Denmark the resolution is 5.6 km x 5.6 km. As already mentioned in the text (section 3.1.2) the formulation by Ho et al., 2006 is used to calculate the air-sea CO2 fluxes, as this has been found to match the EC measurements made at Roskilde Fjord.

Fig 4: "annual mean values" – Did you compute a running mean for each day of the year? or a trend? *Reply: It has been added to the caption of Fig. 4 that a trend was removed: "The trends have been removed from the time series."*

Fig 6: Are these concentrations at the exact hour or hourly averages? *Reply: These are hourly averages, which will be specified in the caption of Fig. 6.*

Fig A1: Fonts are too small. Caption needs additional information. Which model was used? At what resolution?

*Reply: The fonts size will be increase and the additional information will be added to better explain these model inputs. The resolution is the same as the smallest nest in DEHM which is 5.6 km x 5.6 km.*

**New References in the revised text and in the reply:**

Avitabilie et al., (2016) An integrated pan-tropical biomass map using multiple reference datasets. Global Change Biology, doi: https://doi.org/10.1111/gcb.13139

Bregman, B., Segers, A., Krol, M., Meijer, E., and van Velthoven, P.: On the use of mass-conserving wind fields in chemistry-transport models, Atmos. Chem. Phys., 3, 447–457, 2003.

Cai,W.-J.: Estuarine and Coastal Ocean Carbon Paradox: CO2 Sinks or Sites of Terrestrial Carbon Incineration?, Annu. Rev. Mar. Sci, 3, 123–145, https://doi.org/10.1146/annurev-marine-120709-142723, 2011.

Carslaw, K. S., Lee, L. A., Regayre, L. A., and Johnson, J. S.: Climate models are uncertain, but we can do something about it, EOS, 99, https://doi.org/10.1029/2018E0093757, 2018

Chen, C.-T. A., Huang, T.-H., Chen, Y.-C., Bai, Y., He, X., and Kang, Y.: Air–sea exchanges of CO2 in the world's coastal seas, Biogeosciences, 10, 6509–6544, https://doi.org/10.5194/bg-10-6509-2013, 2013.

Gustafsson, E., Omstedt, A., and Gustafsson, B. G.: The air-water CO2 exchange of a coastal sea - A sensitivity study on factors that influence the absorption and outgassing of CO2 in the Baltic Sea, J. Geophys. Res.-Oceans, 120, 5342–5357, https://doi.org/10.1002/2015JC010832, 2015.

Im, U., Christensen, J. H., Geels, C., Hansen, K. M., Brandt, J., Solazzo, E., Alyuz, U., Balzarini, A., Baro, R., Bellasio, R., Bianconi, R., Bieser, J., Colette, A., Curci, G., Farrow, A., Flemming, J., Fraser, A., Jimenez-Guerrero, P., Kitwiroon, N., Liu, P., Nopmongcol, U., Palacios-Peña, L., Pirovano, G., Pozzoli, L., Prank, M., Rose, R., Sokhi, R., Tuccella, P., Unal, A., Vivanco, M. G., Yarwood, G., Hogrefe, C., and Galmarini, S.: Influence of anthropogenic emissions and boundary conditions on multi-model simulations of major air pollutants over Europe and North America in the framework of AQMEII3, Atmos. Chem. Phys., 18, 8929-8952, https://doi.org/10.5194/acp-18-8929-2018, 2018.

Lansø A. S., Mesoscale modelling of atmospheric CO2 across Denmark, PhD thesis, Aarhus University, Department of Environmental Science, 150 pp, 2016.

Laruelle, G. G., Dürr, H. H., Lauerwald, R., Hartmann, J., Slomp, C. P., Goossens, N., and Regnier, P. A. G.: 15 Global multi-scale segmentation of continental and coastal waters from the watersheds to the continental margins, Hydrology and Earth System Sciences, 17, 2029–2051, https://doi.org/10.5194/hess-17-2029-2013, 2013.

Laruelle, G. G., Düurr, H. H., Slomp, C. P., and Borges, A. V.: Evaluation of sinks and sources of CO2 in the global coastal ocean using a spatially-explicit typology of estuaries and continental shelves, Geophys. Res. Lett., 37, 20 https://doi.org/10.1029/2010GL043691, 2010.

Laruelle, G. G., Lauerwald, R., Pfeil, B., and Regnier, P.: Regionalized global budget of the CO2 exchange at the air-water interface in continental shelf seas, Global Biogeochem. Cy., 28, 1199–1214, https://doi.org/10.1002/2014GB004832, 2014.

Le Toan et al., (2011) The BIOMASS mission: Mapping global forest biomass to better understand the terrestrial carbon cycle. Remote Sensing of Environment, 115, 11, 2850-2860

Lovenduski, N. S. and Bonan, G. B.: Reducing uncertainty in projections of terrestrial carbon uptake, Environmental Research Letters, 12, 2017.

Norman, M., Parampil, S. R., Rutgersson, A., and Sahlée, E.: Influence of coastal upwelling on the air-sea gas exchange of CO2 in a Baltic Sea Basin, Tellus B, 65, https://doi.org/10.3402/tellusb.v65i0.21831, 2013.

Thurner et al., (2014) Carbon stock and density of northern boreal and temperate forests. Global Ecology and Biogeography, 23, 297-310

**Figures**

**Fig 1:** The total monthly fluxes from the 7 landcover classifications and the fluxes from the marine areas surrounding Denmark. The marine area has been divided into the North Sea and the Danish inner waters which includes the fjords, the Danish inner straits and Kattegat.

**Simulating the atmospheric CO2 concentration across the heterogeneous landscape of Denmark using a coupled atmosphere-biosphere mesoscale model system**

Anne Sofie Lansø1,a, Thomas Luke Smallman2,3, Jesper Heile Christensen1,4, Mathew Williams2,3, Kim Pilegaard5, Lise-Lotte Sørensen4, and Camilla Geels1

1Department of Environmental Science, Aarhus University, Frederiksborgvej 399, 4000 Roskilde, Denmark

2School of GeoSciences, University of Edinburgh, Edinburgh, EH9 3JN, UK

3National Centre for Earth Observation, University of Edinburgh, Edinburgh, EH9 3JN, UK

4Arctic Research Centre (ARC), Department of Bioscience, Aarhus University, Ny Munkegade 114, 8000 Aarhus, Denmark 5Department of Environmental Engineering, Technical University of Denmark (DTU), Bygningstorvet 115, 2800 Kgs. Lyngby, Denmark

aNow at: Laboratoire des Sciences du Climat et l'Environnement, LSCE/IPSL, CEA-CNRS-UVSQ, Université Paris-Saclay, 91191 Gif-sur-Yvette, France

Correspondence to: Anne Sofie Lansø (anne-sofie.lanso@lsce.ipsl.fr)

Abstract. Surface heterogeneity can be challenging to fully encompass by modelling studies of Although coastal regions only amount to 7% of the global oceans, their contribution to the global oceanic air-sea  $CO_2$  surface exchanges, especially when it comes to land-sea boarders. The relative importance of the marine and the exchange is in-proportionally larger with fluxes in some estuaries being similar in magnitude to terrestrial surface fluxes on the atmospheric concentration were examined by

5 developing a mesoscale modelling framework capable of simulating surface exchanges at a high spatiotemporal resolution. This study exploits the complexity of the Danish landscape and the many land-sea boarders found along the nation's 7,300 km of coastline.  $of CO_{2,2}$

An atmospheric transport model, DEHM, with a horizontal spatial resolution of  $5.6 \text{ km} \times 5.6 \text{ km}$  constituted the basis of the modelling framework. A mechanistic biosphere model, SPA, was coupled to DEHM in order to simulate terrestrial surface

10 exchanges applying a tiling approach with the seven most dominant land-use classes in Denmark to account for sub-grid heterogeneity. Detailed surface fields of pAcross a heterogeneous surface consisting of a coastal marginal sea with esturine properties and varied land mosaics, the surface fluxes of CO2 were used to simulate the air-sea from both marine areas and terrestrial surfaces were investigated in this study together with their impact in atmospheric CO2 exchange for the study region. Monthly mean diurnal cycles of surface water pwere imposed onto these, in order to include short-term variability in surface

The Danish biospheric fluxes simulated by the SPA-DEHM model system concentrations by the usage of a high-resolution modelling framework. The simulated terrestrial fluxes across the study region of Denmark experienced an east-west gradient corresponding to the distribution of the land-use classes and landcover classification, their biological activity . The relative importance of the seven land-use classes varied throughout the year according to their individual growth patterns. A major

20 contribution to the monthly net ecosystem exchange (NEE) through all seasons came from grasslands, while the influence

15 water p.

from croplands increased from March to July. Grasslands had, on an annual basis, the largest impact on the biospheric net uptake with and the urbanized areas. Annually, the Danish terrestrial surface had an uptake of approximately -7000 GgC yr-1. While the marine fluxes from the North Sea and the Danish inner waters annually were smaller with about -1,423-800 GgC yr-1.

5 The total Danish biospheric uptake for 2011 was -6,302 and -1,300 GgC yr-1. Relating the annual natural biospheric surface fluxes to the emitted by fossil fuel combustions and industrial processes by Denmark, the Danish terrestrial uptake corresponded to 52 % of these, while the Danish annual marine uptake was negligible in comparison, although hiding larger seasonal variations.

During 2013-2014, the simulated, their sizes are comparable to annual terrestrial fluxes from individual landcover classifications

- 10 in the study region, and hence not negligible. The contribution of terrestrial surfaces fluxes were easily detectable on both simulated and measured concentrations of atmospheric  $CO_2$  concentrations compared well with measurements made at the Risøtall tower located on the shore of Roskilde Fjord (R = 0.88 and RMSE = 4.87 ppm). The origin of the simulated concentrations at Risøvaried between seasons with biospheric fluxes and fossil fuel emissions having the largest impact on the variations. Impact from Roskilde fjordwas difficult to detect at the only tall tower site in the study region. Although the
- 15 tower is positioned next to Roskilde fjord, the local marine impact was not distinguishable in the simulated concentrations. These difficulties in simulating the local concentrations. But the regional impact from the Roskilde Fjord might arise from (*i*) the fjord not being adequately resolved in the constructed model system, (*ii*) the lack of a realistic representation of the surface water pdynamics, or (*iii*) that the fjord is not in the simulated footprint and only had a modest impact on the simulated atmospheric at the Risøtall towerDanish inner waters and the Baltic sea increased the atmospheric concentration by up to 0.5
- 20 ppm during the winter months.

**Copyright statement. TEXT**

To have the best chance of accurately predicting the future evolution of the carbon cycle, and its implications for our climate, it is important to eliminate, or at least minimise, the uncertainty that exists in the present estimates. Enhanced knowledge, improved temporal resolution of surface exchange processes and improvements in the spatial surface representation are factors

25 that could minimise these uncertainties.-

On an annual basis the biosphere is estimated to absorb

**1** Introduction**

Understanding the natural processes responsible for absorbing just over half of the anthropogenic carbon emitted to the atmosphere, will help decipher future climatic pathways. During the last decade, the ocean and the biosphere are annually

30 estimated to take up  $2.4 \pm 0.5$  PgC yr-1 and  $3.0 \pm 0.8$  PgC yr-1 of the  $9.4 \pm 0.5$  PgC yr-1 anthropogenic carbon emitted to the atmosphere (Le Quéré et al., 2018). The heterogeneity and the dynamics of the land surface complicates such estimates.

Biosphere models of various complexity have been developed to spatially simulate surface fluxes of  $CO_2$ , but global bottom-up estimates are poorly constrained by surface observation. Although a huge observational effort has been and is still being put into observing surface exchanges, the observations are far from a global surface representation (Ciais et al., 2014; Zscheischler et

- 5
- In order to omit the lack of surface observation, atmospheric models have been used to construct inverse modelling systems, where atmospheric measurements have been used to constrain surface fluxes (e. g. Baker et al. (2006); Gurney et al. (2002, 2004); Peylin et ). These atmospheric inversions are capable of capturing the year to year changes in natural surface fluxes, the magnitude and distribution of regional fluxes, and distinguish between land and ocean fluxes (Le Quéré et al., 2015). However, atmospheric inversions are limited by the availability of atmospheric measurements, and therefore, only regional inversions have been
- 10 conducted in areas with very dense data network, e.g, Europe (Broquet et al., 2011; Rödenbeck et al., 2009), the corn belt of the United States (Lauvaux et al., 2012), and national scale inversions for the Netherlands (Meesters et al., 2012; Tolk et al., 2011) but future estimates of the land uptake are bound with large uncertainties (Friedlingstein et al., 2014) that can be attributed to model structural uncertainties, uncertain observations and lack of model bench-marking (Cox et al., 2013; Luo et al., 2012; Lovenduski and . To have the best chance of accurately predicting the future evolution of the carbon cycle, and its implications for our
- 15 climate, it is important to minimise the uncertainties that exist presently (Carslaw et al., 2018). Enhanced knowledge and a better process understanding in ecological theory and modelling could potentially reduce the model structural uncertainties (Lovenduski and Bonan, 2017) which together with improvements in spatial surface representation could minimise the current uncertainties.

Studying surface exchanges of  $CO_2$  on regional to local scale can be accomplished with mesoscale atmospheric transport

- 20 models. Their resolution is in the range from 2 km to 20 km and their advantage is their capability to get a better processes understanding of both atmospheric and surface exchange mechanisms in order to improve the link between observations and models at all scales, i.e. for both mesoscale, regional and global models (Ahmadov et al., 2007). The higher spatial resolution of mesoscale models allows for a better representation of atmospheric flows and for a more detailed surface description, which in particular for heterogeneous areas is advantageousnecessary. In previous mesoscale model studies, biosphere models have
- been coupled to the mesoscale atmospheric models ranging in their complexity from simple diagnostic (Sarrat et al., 2007b; Ahmadov et al., 2007, 2009) to mechanistic process based biosphere models (Tolk et al., 2009; Ter Maat et al., 2010; Smallman et al., 2014; Uebel et al., 2017). The modelled  $CO_2$  concentrations and surface fluxes from mesoscale model systems compare better with observations than global model systems (Ahmadov et al., 2009). The atmospheric impact on surface processes related to the ecosystem's sensitivity and  $CO_2$  exchange can be examined in greater details (Tolk et al., 2009) and tall towers
- 30 footprints can be studied more concisely (Smallman et al., 2014).

The heterogeneity of coastal ocean contributes with a large uncertainty to assessment of the Heterogeneity can also be considerable in coastal oceans, and like terrestrial surface fluxes, the high spatiotemporal variability leads to large uncertainties in estimates of coastal air-sea  $CO_2$  exchange (Regnier et al., 2013)fluxes (Cai, 2011; Laruelle et al., 2013). Coastal seas , rich in nutrient and organic material, might contribute with an in-proportional amount to the global play an important role in

35 the carbon cycle facilitating lateral transport of carbon from land to open oceans, but almost 20 % of the carbon entering

estuaries are released to the atmosphere, while 17 % of the carbon inputs to coastal shelves comes from atmospheric exchange (Regnier et al., 2013). The air-sea  $CO_2$  flux, with regards to its limited spatial area when compared to the open oceans (Gattuso et al., 1998). exchange is in general numerical larger for estuaries than shelf seas (Chen et al., 2013; Laruelle et al., 2010, 2014), and can for estuaries annually be as large as 1,958 gC m-2 yr-1, while continental shelf seas have fluxes in the range of -154

- 5 gC/m2/yr to 180 gC m-2 yr-1 (Chen et al., 2013). The large spatial and temporal heterogeneity of coastal ocean adds to the large uncertainty related to the annual estimates of the air-sea CO2 exchange (Regnier et al., 2013). The observed high spatial and temporal variability (Kuss et al., 2006; Leinweber et al., 2009; Vandemark et al., 2011; Norman et al., 2013; Mørk et al., 2016) are not always included in marine models (Omstedt et al., 2009; Gypens et al., 2011; Kuznetsov and Neumann, 2013; Gustafsson et al., 2015; Valsala and Murtugudde, 2015), let alone taken into account in atmospheric mesoscale systems
- 10 simulating CO2 (Sarrat et al., 2007a; Geels et al., 2007; Law et al., 2008; Tolk et al., 2009; Broquet et al., 2011; Kretschmer et al., 2014). Moreover, But a recent study has found that short-term variability in the partial pressure of surface water CO2 (pCO2) can be very influential of the annual flux for some coastal areas (Lansø et al., 2017). substantially can affect simulated annual fluxes in certain coastal areas in their case the Baltic Sea (Lansø et al., 2017). Moreover, direct Eddy Covariance (EC) measurements in the Baltic Sea have shown that upwelling events with rapid changes of pCO2 greatly increase the air-sea CO2
- 15 exchange (Kuss et al., 2006; Rutgersson et al., 2009; Norman et al., 2013)

In this study we aim to simulate surface exchanges of  $CO_2$  at a high spatial-temporal resolution across a region, neighboring the Baltic Sea, alternating between land and coastal sea together with mesoscale atmospheric transport. Interactions between the atmosphere – ocean, and atmosphere – biosphere are contained in a mesoscale modelling framework together with high resolution anthropogenic emissions of A newly developed mesoscale modelling system is used to assess and understand the

- 20 dynamics and relative importance of the marine and terrestrial  $CO_2$  fluxes. The Danish Eulerian Hemispheric Model (DEHM) forms the basis of the framework, while the mechanistic biospheric Soil-Plant-Atmosphere model (SPA) is dynamically coupled to the atmospheric model. Both models are driven by methodological data from the Weather Research and Forecasting model (WRF). The air-sea  $CO_2$  exchange is simulated at a high temporal resolution with the best applicable surface fields of  $pCO_2$ for the Danish marine areas. Tall tower observations are used to evaluate the simulated atmospheric concentrations of  $CO_2$ .
- 25 Section 2 is dedicated to describe the study region, which is followed by a detailed description and evaluate of the atmospheric and biospheric model components of the developed model system in Sect. 3. Section 4 contains results, while discussion and conclusion follow in Sect. 5 and 6.

**2 Study area**

The study area comprises of Denmark, a country that is characterised by a mainland (Jutland) and many smaller islands, all containing varied land mosaic <del>containing of</del> urban, forest and agricultural areas. With more than 7,300 km of coastline encircling approximately 43,000 km2 of land, many land-sea borders are found throughout the country adding to the complexity <del>. The developed mesoscale modelling system is used to assess and understand the dynamics and relative importance of (Fig. 1). Denmark is positioned in the transition zone between the Baltic Sea, a marginal coastal sea with low salinity, and</del> the marine and terrestrial North Sea, a continental shelf sea. Bordering the Baltic Sea, the Danish inner waters are rich on nutrients and organic material (Kuliński and Pempkowiak, 2011). This fosters high biological activity in spring and summer lowering surface water  $pCO_2$  fluxes across this particular region with a special focus on the impact from the Roskilde Fjord system. The Danish Eulerian Hemispheric Model, DEHM, forms the basis of the framework, while the mechanistic biospherie

- 5 Soil-Plant-Atmosphere model, SPA, is dynamically coupled to the atmospheric model. The allowing for uptake of atmospheric CO2. In winter, mineralisation increases pCO2 (Wesslander et al., 2010), and outgassing of CO2 to the atmosphere takes place. The North Sea is a persistent sink of atmospheric CO2, where a continental shelf-sea pump efficiently removes pCO2 from the surface water and transport it to the North Atlantic Ocean (Thomas et al., 2004). This study uses definition of the Danish exclusive economic zone (EEZ) to estimate the Danish air-sea CO2 exchange is simulated at a high temporal resolution
- 10 with the best applicable surface fields of pexchange, as the coastal state (in this case Denmark) has the right to explore, exploit and manage all resources found within its EEZ (United Nations Chapter XXI: Law of the Sea, 1984). The Danish EEZ is approximately 105,000 km2.

A tiling approach with the seven most common biospheric landcover classification were selected for the current study including deciduous forest (3,348 km2), evergreen forest (1,870 km2), winter barley (1,211 km2), winter wheat and other

[revised manuscript text omitted]

The necessary meteorological parameters for DEHM were simulated by the Weather Research and Forecast Model (WRF) (Skamarock et al., 2008), nudged by six hourly ERA-Interim meteorology, which was also used as initial and boundary conditions (Dee et al., 2011).

**3.1.2 Air-sea**

20

30

**The-**

**3.1.2 Air-sea CO2 exchange**

The exchange of CO2 between the atmosphere and the ocean,  $F_{CO_2}$ , was calculated by  $F_{CO_2} = Kk_{660}\Delta pCO_2 F_{CO_2} = Kk_{660}\Delta pCO_2$ , where K is solubility of CO2 calculated as in Weiss (1974),  $k_{660}$  is the transfer velocity of CO2 normalised to a Schmidt num-

[revised manuscript text omitted]
 experiences phenological problems with rapid leaf growth in response to environmental drivers, which causes a delay in spring photosynthesis. lacks a labile / non-structural carbohydrate store needed to driver rapid leaf expansion with the onset of spring;
- 10 instead leaf expansion is dependent on available photosynthate on a given time step. Therefore, SPA's LAI is lower early in the growing season resulting in a biased slow photosynthetic activity and an underestimate in the magnitude of NEE as seen at Gludsted (Fig. 4). Voulund alternates between winter and spring barley for the calibration period starting with winter barley in 2009. Note that the whole observed time series of NEE at Voulund is shown together with model NEE of both winter and spring barley (Fig. 24c and d). While the phenology and amplitude are well captured by for spring barley at Voulund, SPA is
- 15 not able to capture the seasonal amplitude of the winter barley that seems to be more sensitive to the meteorological drives, and seasons with harder winters had lower NEE peaks in summer (winter 2010-11, 2012-13). At the grassland site Skjern Enge, NEE is for winter, spring, and the first part of the summer reasonably modelled. The difficulties for late summer and autumn arise from the management practises at the site, where both grazing and grass cutting are conducted, limiting NEE (Herbst et al., 2013). Although grazing is included in the SPA model, it does not simulate the same reduction in NEE.
- Examining the performance of SPA at a higher temporal resolution (Table 3), the correlations are better for hourly values than daily for the landcover classification having problems with the phenology (evergreen forest and winter barley), because SPA is capable of reproducing the diurnal variability. For the remaining landcover classifications  $R^2$  and RMSE are improved when going from hourly to monthly averages of NEE. Zooming in on shorter time windows, the timing of the diurnal cycle are in accordance with measured NEE (see e.g. Fig S9), but the amplitude is underestimated by SPA.

**25 3.3 Model coupling**

In the inner most nest of DEHM for the area of Denmark, a coupling was made between DEHM and SPA. Thus, the coarser optimized biospheric fluxes from CT2015 were for Denmark replaced by hourly SPA simulated fluxes. With this change, the spatial resolution for the biosphere fluxes was increased from  $1^{\circ} \times 1^{\circ}$  to 5.6 km  $\times$  5.6 km allowing for a better representation of the Danish surface, and hence also the biospheric fluxes.

30 A tiling approach with seven most common biospheric land-use classification were selected for this study including deciduous forest, evergreen forest, winter wheat and other winter crops, winter barley, spring barley and other spring crops, grassland and agricultural other, but excluding urbanised areas. This classification corresponds to the actual crop distribution of 2011 (Jepsen and Levin, 2013). Denmark is dominated by agriculture, and more than 60 % of the used classification is agricultural

land. DEHM provides on an hourly basis meteorological drivers and atmospheric concentrations to SPA, while SPA each hour returns NEEto DEHM.

**3.3 Observations of atmospheric-**

Tall tower continuous measurements of atmospheric concentrations made by a Picarro 118 m above the surface have been

5 conducted at the Risøsite since the middle of 2013. The Risøsite is located on the eastern inner shore of Roskilde Fjord (55°41'N, 12°05'E), Zealand, which is a narrow microtidal estuary 40 km long with a surface area of 123 km2 and a mean depth of 3 m (Mørk et al., 2016).

**4 Results**

The model system was run from 2008 to 2014, however, the first 3 years were with the first three years regarded as a spin-

10 up period. Given that the land-use classification corresponds to the actual distribution of 2011 an emphasis will be put on the terrestrial fluxes for this particular year during the analysis together with an estimation of the annual Danish carbon budget. Measurements of In the following sections the terrestrial and marine surface fluxes will be presented first, followed by measurements of atmospheric  $CO_2$  from Risø campus tower that will be used to assess the performance of the DEHM-SPA model system, and evaluate local impacts from fjord systems on atmospheric  $CO_2$  concentrations.

**15 4.1 Surface fluxes**

**4.1.1 Biospheric fluxes**

Across Denmark, there is As shown in Fig. 5, the SPA model simulates an east-west gradient in the simulated biospheric fluxes of carbon-NEE for both January and July in 2011. Larger fluxes of gross primary productivity (GPP), ecosystem respiration and NEE are evident values of NEE are found in the western part of Denmark, while the islands and Eastern Jutland have lower

20 biosphere fluxes(Fig. 5)... This gradient follows the distribution of the individual land-use classifications (Appendix A Fig Allandcover classifications (Supplement Fig. S8)and their grow patterns, their phenology and productivity, but also reflects the population density which is highest urbanisation which is denser in the eastern part of the country.

Table 2 and Table 3 show the contribution of each individual land-use class to the total GPP and respiration on a national scale (see Table B1 to Table B3 Appendix B for monthly GPP, respiration and NEE per land-use class). The monthly contributions to

- 25 the country-wide total inherently reflect the total area for each land-use class. In winter, GPP is highest for evergreen, grassland and agricultural other. These land-use classes are well represented in Western Jutland explaining why the largest biological production is found here during January During January, total ecosystem respiration dominates NEE. Evergreen forests and grasslands are well represented in Western Jutland and even though these landcover classes have GPP, they are still dominated by total ecosystem respiration, but their total ecosystem respiration can be higher than the other landcover classifications
- 30 because of the contribution from the autotrophic respiration that in SPA depends on GPP. During July, the productivity is at

its highest for all landcover classes dominating total ecosystem respiration resulting in negative NEE (Fig. 5a). As the crops develop, their contribution to the total monthly GPP increases, and in July winter crops are together with grasslands the most productive land-use classes. Also, deciduous forest increases its share of GPP from the onset of the growing season, but as its spatial extent only amount to 10 %, its monthly contribution is never dominant. Respiration is less concentrated for individual

5 land-use classes and the individual monthly contributions vary much less for respiration than GPP throughout the year, though with a notable increase in contribution from both winter and spring crop through spring and summer until harvest (Table 3). The highest contributions of respiration are throughout the year found for grassland and agriculture other. 6), and the gradient across the country is more likely a result of the urbanization.

Figure 6 shows the average monthly contribution from each landcover classification to the country wide NEE, which

- 10 inherently follows their productivity but also reflects the area covered by each landcove type with highest peaks for winter wheat and grasslands during June. During winter, the spread amongst the landcover classifications are smaller, but still with numerically larger monthly fluxes for the landcover classifications with largest area. Integrating over all land-use classes for 2011landcover classification, the Danish terrestrial land surfaces is a net source of CO2 to the atmosphere in the months from January October to April, and October to December, with the highest monthly release of 1.129 GgC monthyr063±154 GgC
- 15 month-1 in December. From May to September, the biosphere is a net sink with a maximum uptake in June of -4,687 GgC monthyr982±385 GgC month-1. The total annual surface exchange of CO2 between the atmosphere and Danish biosphere is  $-6,302-7,337\pm1,468$  GgC yr-1 for 2011, where grassland, where winter wheat has the largest contribution with -1,423-2,342  $\pm 1,045$  GgC yr-1.

**4.1.2 Marine fluxes**

- 20 The air-sea CO2 exchange in the Danish inner waters experiences experience large seasonal variations, while the variations in the North Sea are less pronounces as illustrated by Fig. 7. Bordering the Baltic Sea, The minerlisation in winter increases the surface water pCO2 in the Danish inner waters are rich on nutrients and organic material (Kuliński and Pempkowiak, 2011) . This fosters high biological activity in spring and summer lowering surface water pallowing for uptake of atmospheric . In winter, mineralisation increases p(Wesslander et al., 2010), and resulting in outgassing of CO2 to the atmospheretakes place.
- 25 The North Sea is a continuously sink of atmospheric , where a continental shelf-sea pump removes p, while uptake occurs during spring and summer months following the decrease in surface water  $pCO_2$  from the surface water and transport it to the North Atlantic Ocean (Thomas et al., 2004).

To estimate the annual exchange of between the atmosphere and the ocean for the Danish marine areas, the exclusive economic zone (EEZ) is used. In the EEZ the coastal state (in this case Denmark) has the right to explore, exploit and manage

30 all resources found within it (United Nations Chapter XXI: Law of the Sea, 1984). Thus, assuming the air-sea exchange counts as a natural resource for Denmark, the air-sea flux from EEZ was used for the annual budget estimation of . The 2011 due to biological activities.

The simulated annual air-sea  $CO_2$  exchange in the 105,000 km2 covered by the Danish EEZ amounts to -422 GgC yr-1. However, this number hides masks large spatial differences and monthly numerical larger fluxes (Fig.6). While the North Sea area contained within the EEZ continuously had monthly uptakes in the range -73 GgC mon-1 to -191 GgC mon-1 with an annual accumulation of 1,765 GgC yr-1, the monthly fluxes from the near coastal (marine areas extending up to 10 km of shore) Danish inner waters varied in the range -46 GgC mon-1 to 540 GgC mon-1 annually releasing 1,343 GgC yr-1 to the atmosphere.

**5 4.1.3 Danish budget**

The annual budget for year 2011 is assessed, because the crop distribution used for the land-use classification was based on data from this specific year as was the spatial distribution of the Danish fossil fuel emissions. In 2011, Denmark emitted 12,205 GgC yr-1 of to the atmosphere due to fossil fuel combustion and industrial processes (Nielsen et al., 2015). The Danish terrestrial biosphere took up -6,302 GgC yr-1 of during 2011, which equals 52 % of the emitted by anthropogenic activity in Denmark. The marine uptake of -422 GgC yr-1 is annually less influential and corresponds to only 3.5 % of the anthropogenic emitted .

10

**4.2 Atmospheric CO2 concentrations**

The time series of measured and simulated CO2 show good agreements (Fig. 8) with  $\frac{R}{R^2} = \frac{0.88 \cdot 0.77}{2}$  and RMSE = 4.87 ppm for daily averaged time series demonstrating that the model is capable of capturing the synoptic scale variability. Also, good statistical measures are obtained for the hourly time series with  $\frac{R}{R^2} = \frac{0.84 \cdot 0.71}{2}$  and RMSE = 5.95 ppm, but the short-term variability is was not always fully captured by the model. Overall the model simulates the All in all the evaluation shows that

- 15 variability is was not always fully captured by the model. Overall the model simulates the All in all the evaluation shows that the model can capture the overall variability of the atmospheric  $CO_2$  quite well, indicating that the simulated surface exchange of concentrations and fluxes. Moreover, the higher resolution in both transport model and surface fluxes results in a better model performance in simulating atmospheric  $CO_2$  is acceptable. concentrations (Fig. S10).
- To investigate the origin of the CO2 simulated at the Risø site, concentration rose plots of simulated atmospheric CO2 have been made (Fig. 9). The concentration rose shows the wind direction and associated CO2 concentrations. Roskilde Fjord lies in the approximate sector of 200°- 360° relative to the Risøtower, the city of Roskilde (with 50,000 inhabitants) is positioned approximately 5 km south to south-west of the tower, while Copenhagen lies 20 km east. Division has been made between seasons, and day and night time values both showing distinct seasonal and diurnal patterns. The highest values of CO2 are obtained during winter, where very little diurnal variation is seen. During summer the lowest values are obtained in particular 25 during daylight, when photosynthesis occurs.

The individual contribution from fossil fuel emissions, marine and biospheric exchanges to the atmospheric  $CO_2$  (see Appendix C-Supplement Fig. S11 to Fig. \_\_S13) indicate that the biosphere contributes most to the variations simulated at RisøRisø (Fig. S12) -\_\_ both seasonally and daily. Emissions of fossil fuel experience little diurnal variability, but seasonally with the greatest contribution during autumn and winter (Fig. S11). Highest values are seen originating from the sectors encapsulating the city of Roskilde and the capital region. In all seasons, the simulated oceanic contribution is negative, i.e. indicating uptake of atmospheric  $CO_2$ , but the marine contribution is small with little variation (Fig. S13). The less negative values in autumn and winter may be a result of the simulated outgassing of  $CO_2$  from the Baltic Sea and Danish inner waters during the winter season (Lansø et al., 2015), which however is still dominated by the uptake by global open oceans. The local impact from Roskilde Fjord is difficult to detect in the marine concentration plots. Flux measurements at Roskilde Fjord have shown uptake of  $CO_2$  during spring, while release in the remaining seasons (Mørk et al., 2016), which is accurately captured by the modelling system (Lansø et al., 2017). A footprint analysis of the **RisøRisø** tower has shown that the fluxes from Roskilde Fjord has have a contribution to the total  $CO_2$  flux measured at the top of the 118 m high tower, but only minor,

- 5 since fluxes over water typically is are an order of magnitude smaller than fluxes over land (Sogachev and Dellwik, 2017). Therefore, we investigated a period with observed large outgassing from Roskilde Fjord a storm event in October 2013 that was observed to increase the monthly release of  $CO_2$  in the fjord by 66 % (Mørk et al., 2016). The storm event passed Denmark on 28 October 2013, and at 06 UTC southerly winds transport air masses with higher  $CO_2$  towards the RisøRisø site (Fig. 10a), while at the same time a detectable increase in the oceanic contribution to the  $CO_2$  concentration at the Roskilde Fjord system
- 10 is seen (Fig. 10b). The model system simulates the small peak in the observed atmospheric CO2 concentrations for 28 October (Fig. 11a) at the **RisøRisø** site, but distinguishing between contributions from fossil fuel emissions, the biosphere and the ocean to the atmospheric CO2 concentration at **RisøRisø** (Fig. 11b) reveals no oceanic impact, and hence no apparent influence from Roskilde Fjord during the storm event.

**5 Discussion**

20

**15 5.1 Biospheric-Surface fluxes**

Spatially, the Danish biospheric fluxes of follows the land-use classification that mirrors the population density being highest in eastern and lowest in western Denmark. While GPP follows phenology and productivity resulting in changing percentage contribution from the land-use classes to the monthly total GPP, the mutual proportions of respirations are much less varied. Autotropic respiration depends on plant productivity, but the heterotropic respiration is temperature dependent and will change proportionally for each land-use class maintaining a more constant percentage-wise distribution.

- The simulated annual uptake by deciduous forest of  $-300-284 \pm 21$  gC m-2 yr-1 for 2011, fits the the period 2011-2014 is within the observed range of annual estimated NEE at Sorø from 1996 to 2009 spanning 32 gC m-2 yr-1 to -331 gC m-2 yr-1 (Pilegaard et al., 2011). Even though SPA experiences a lag in the seasonal onset for the evergreen forest, the annual estimated uptake of -386 gC m-2 yr-1 compares well with previous estimates of temperate evergreen forests with -402 gC m-2 yr-1
- 25 (Luyssaert et al., 2007) and Danish evergreen plantations of -503 gC m-2 yr-1 (Herbst et al., 2011). However, improvements Improvements to the evergreen plant functional type in SPA are needed, and an addition of a labile pool to the evergreen carbon assimilation would omit the seasonal lag (Williams et al., 2005). Such adjustments have already been made to the DALEC carbon assimilation system utilised by SPA (Smallman et al., 2017) substantially improving the representation of terrestrial phenology, but not yet incorporated into SPA. The annual estimated uptake of -355 ± 41 gC m-2 yr-1 is in the low range
- 30 of previous estimates of temperate evergreen forests with -402 gC m-2 yr-1 (Luyssaert et al., 2007) and Danish evergreen plantations of -503 gC m-2 yr-1 (Herbst et al., 2011). This could be caused by the slow leaf onset in spring, inhibiting the productivity at the beginning of the growing season.

Previous annual estimates at Danish agricultural field sites found carbon uptake with of -31 gC m-2 yr-1 estimated from a mixed agricultural landscape (Soegaard et al., 2003) and -245 gC m-2 yr-1 from winter barley at a winter barley site (Herbst et al., 2011). The SPA-DEHM model system simulated annual uptakes for winter wheat and spring crops of -137 of -252  $\pm$  113 gC m-2 yr-1 and -207, and spring crops of -179  $\pm$  28 gC m-2 yr-1, respectively, and thus fits the previous estimates well,

- 5 while winter barley had a small release of 32 smaller uptake of  $-82 \pm 91$  gC m-2 yr-1 with large standard deviation potentially resulting in small annual releases. The calibration and validation (Fig. 4c) shows difficulties in simulating the observed NEE during growing seasons for winter barley particularly after cold and snow covered winters(winter 2010-11, 2012-13). For 2011 winter barley had the smallest total area of the agricultural land-use classifications, and was dominated by the respiration. As pointed out in previous studies, the crop modelling component in SPA could likewise be improved e.g. by inclusion of
- 10 intra-seasonal crops (Smallman et al., 2014).

The current study annually estimated the Danish grasslands to be a sink of  $CO_2$  with  $-205-210 \pm 43$  gC m-2 yr-1, which are similar to albeit slightly smaller than the -267 gC m-2 yr-1 observed at the Skjern Enge grassland site (Herbst et al., 2011) during 2009-2011 (Herbst et al., 2013) and the -312 gC m-2 yr-1 observed at the Lille Valby grassland site, Denmark (Gilmanov et al., 2007). The European grassland study by Gilmanov et al. (2007) found large variation in annual fluxes from grassland

- 15 driven by environmental conditions and management practises at the sites varying from 171 gC m-2 yr-1 to -707 gC m-2 yr-1, but with most site having an annual uptake of carbon. As seen in Fig. 4e more work on grassland calibration could have been done, but the conditions and management regimes at Skjern Enge does not necessarily fit the rest of the Danish grasslandgrasslands. With the chosen parameters, very comparable results were obtained indicating that such an additional calibration might not be advantageous.
- A tilling approach has been used for the land-use-landcover classification in the SPA-DEHM modelling framework, including sub-grid heterogeneity in the model system. However, the seven land-use landcover classes do not fully encompass the ecosystem variability in Denmark. Both grassland and agricultural other cover a broad range of sub-categories with both heather and meadow included in the grassland class, while agricultural other among other things e.g. contains vegetables fields, hedgerows, woodland patches and uncultivated land highlighting the need to adopt approaches allowing for generating novel
- 25 spatially varying parameter sets (Bloom et al., 2016). Moreover, large urbanised areas are not accounted for in the current classes either. Adding more land-use landcover classifications could give a better and more realistic surface description, if data for both calibration of and validation for the lacking land-use landcover classes, preferably from similar climatic region as Denmark, were available.

**5.2 National budget**

30 The current estimate of the global carbon budget appraises the global biosphere to take up 32% of the emitted by fossil fuel and industry, while the global oceans are estimated to take up 26% (Le Quéré et al., 2018). National scale biospheric uptakes can vary greatly between countries depending on land coverage, land-use and management practices (Janssens et al., 2005). Meesters et al. (2012) estimated an annual biospheric uptake for the Netherlands of -17,400 GgC Compatible marine fluxes to previous estimates are obtained for the study region. On an annual basis, the Danish inner waters were found to be a source of

 $30 \text{ gC m}^{-2} \text{ yr}^{-1}$ , which is agreeing with most previous studies. Wesslander et al. (2010) estimated Kattegat to act as a small sink of -14 gC m-2 yr-1 (approximately -497 based on measurements of water chemistry, while Norman et al. (2013) on the contrary found a release of 19 gC m-2 yr-1), which corresponded to 33% of the annual fossil fuel emitted by the country. An annual uptake of -99 using a biogeochemical model of the Baltic Sea. Measurements from Danish fjords on the other

- 5 hand consistently point towards these marine areas being annual sources of  $CO_2$  with values in the range of 41 gC m-2 yr-1 has been estimated for Scotland by Smallman et al. (2014)to 104 gC m-2 yr-1 (Gazeau et al., 2005; Mørk et al., 2016). The current study estimates a total annual biospheric uptake of -6,302 Gg C the North Sea to be a sink of -29 gC m-2 yr-1, which equals 52 % of the emitted by anthropogenic activities in Denmark. Integrating over all land-use classes the uptake per area is -195 gC is very close to previous estimates, both measured and modelled, of -20 m-2 yr-1, which places this Danish estimate
- 10 within the bounds of the previous national estimates at similar latitudes. Caution should be taken when assessing the Danish budget. We present a one year snap-short of the state of the Danish surface exchanges of , but important processes such as product use, biomass burning and river runoff linking land and ocean are lacking in our estimate in order to fully close the budget. and -25 m-2 yr-1 (Thomas et al., 2004; Prowe et al., 2009).

**5.2 Atmospheric CO2 and land-sea signals**

- 15 The WRF are in general capable of simulating the observed wind patterns, while the the overestimation of the wind velocity could lead to an overestimation of the atmospheric mixing. However, the SPA-DEHM modelling system resembles the synoptic and